# The impact of temperature inversions on black carbon and particle mass concentrations in a mountainous area

Kristina Glojek[1], Griša Močnik[2,7], Honey Dawn C. Alas[3], Andrea Cuesta-Mosquera[3], Luka Drinovec[2,7], Asta Gregorič[4,7], Matej Ogrin[1], Kay Weinhold[3], Irena Ježek[4], Thomas Müller[3], Martin Rigler[4], Maja Remškar[2], Dominik van Pinxteren[3], Hartmut Herrmann[3], Martina Ristorini[5], Maik Merkel[3], Miha Markelj[6] and Alfred Wiedensohler[3]

[1]Department of Geography, Faculty of Arts, University of Ljubljana, Ljubljana, 1000, Slovenia
[2]Department of Condensed Matter Physics, Jozef Stefan Institute, Ljubljana, 1000, Slovenia
[3]Leibniz Institute for Tropospheric Research, Leipzig, 04318, Germany
[4]Aerosol d.o.o., Ljubljana, 1000, Slovenia
[5]Department of Bioscience and Territory, University of Molise, Pesche, 86090, Italy
[6]Škovine 4, Žlezniki, 4228, Slovenia
[7]Center for Atmospheric Research, University of Nova Gorica, Ajdovščina, 5270, Slovenia

*Correspondence to*: Kristina Glojek (k.glojek@gmail.com)

**Abstract.** Residential wood combustion is a widespread practice in Europe with a serious impact on air quality, especially in mountainous areas. While there is a significant number of studies conducted in deep urbanized valleys and basins, little is known about the air pollution processes in rural shallow hollows where around 30 % of the people in mountainous areas across Europe live. We aim to determine the influence of ground temperature inversions on wood combustion aerosol pollution in hilly, rural areas. The study uses Retje karst hollow (Loški Potok, Slovenia) as representative site for mountainous and hilly rural areas in central and southeastern Europe with residential wood combustion. Sampling with a mobile monitoring platform along the hollow was performed in December 2017 and January 2018. The backpack mobile monitoring platform was used for the determination of equivalent black carbon (eBC) and particulate matter (PM) mass concentrations along the hollow. To assure high quality of mobile measurement data, intercomparisons of mobile instruments with reference instruments were performed at two air quality stations during every run. Our study showed that aerosol pollution events in the relief depression were associated with high local emission intensities originating almost entirely from residential wood burning and shallow temperature inversions (58 m on average). The eBC and PM mass concentrations showed stronger associations with the potential temperature gradient ($R^2 = 0.8$) than with any other meteorological parameters taken into account (ambient temperature, relative humidity, wind speed, wind direction and precipitation). The strong association between the potential temperature gradient and pollutant concentrations suggests that even a small number of emission sources (total 243 households in the studied hollow) in similar hilly and mountainous rural areas with frequent temperature inversions can significantly increase the levels of eBC and PM, and deteriorate local air quality. During temperature inversions the measured mean eBC and $PM_{2.5}$ mass concentrations in the whole hollow were as high as $4.5 \pm 2.6$ µg m$^{-3}$ and $48.0 \pm 27.7$ µg m$^{-3}$, respectively, which is comparable to larger European urban centres.

## 1 Introduction

One of the biggest emission sources of Particulate Matter (PM) and carbonaceous aerosols is wood burning (Van Der Werf et al., 2010). It is a worldwide problem affecting undeveloped and developed countries as well as rural areas and cities (e.g. Bonjour et al., 2013, Bond et al., 2004; Yttri et al., 2005; Caseiro et al., 2009; Fuller et al., 2014; Puxbaum et al., 2007; Stohl et al., 2007; Reis et al., 2019). There is a major concern regarding the impact of PM particles from wood burning on health (e.g. Kocbach Bølling et al., 2009; Fong and Nussbaumer, 2012) and climate (Bond et al., 2013).

Wood combustion represents 51 % of the total $PM_{2.5}$ emissions in the European Union (EU), which is the highest among all emission sources (European Environmental Agency, 2020). Not only is wood used as a heating fuel in the Nordic and Alpine regions (Puxbaum et al., 2007; Yttri et al., 2005; Herich et al., 2014 and the reference therein), but is becoming a more common practice elsewhere, also in the Mediterranean region (Titos et al., 2017 and the reference therein). Fuller et al. (2014) state that promotion of biomass as a $CO_2$ neutral fuel, higher taxes in heating diesel and the financial crisis have resulted in the

widespread use of wood as fuel in Europe in the last years. In addition, actions implemented for reduction of PM emissions in Europe mostly focus on road traffic and cities whereas residential wood combustion and rural areas remain underrepresented with only few studies (for instance, Reis et al., 2009; Wählin et al., 2010; Becerril-Valle et al., 2017) showing that wood combustion in rural regions can induce PM exposures tantamount or often even higher than those from traffic.

Populated relief depressions are particularly vulnerable to air pollution as a combined result of high emissions and unfavourable

meteorological conditions for dilution of pollutants. As a result of near-surface radiative cooling, ground temperature inversions frequently and distinctly occur in concave-shaped landforms, especially in the cold season. At the bottom of the surface depressions a strong stable layer is formed preventing vertical mixing of pollutants and ventilation, leading to the accumulation of pollution (Sandradewi et al., 2008; Lyamani et al., 2010). According to Herich et al. (2014) concentrations of carbonaceous aerosols in the Alpine valleys are commonly up to 6 times higher than in urban or rural locations at the foothills

of the Alps.

The problem of air pollution in deep urbanized valleys and basins was recognized and systematically studied in the 20[th] century already (e.g. Whiteman, 1990, 1999 and reference therein). Severe episodes of air pollution and the lack of knowledge about complex atmospheric processes in populated terrain hollows are reasons for several studies of this topic in the last decade as well (e.g. Chazette et al., 2005; Malek et al., 2006; Chuang et al., 2008; de Franceschi and Zardi, 2009; Gohm et al., 2009;

Silcox et al., 2012; Green et al., 2013; Rendón et al., 2015). There is a predominant use of atmospheric models that continue to have difficulties predicting dispersion in the strongly stable boundary layer in complex relief (e.g. Nielsen-Gammon et al., 2010; Ritter et al., 2013; Cuvelier et al., 2014; Tartakovsky, Stern, and Broday 2016; Taylor, Hirsch, and Burns 2017). The problem of models is the use of empirical data collected on a flat, homogenous terrain that cannot be simply transferred to a complex relief where atmospheric conditions change on much shorter distances (Holmes et al., 2015). Consequently, air

pollution problems in populated relief depressions, especially in small-scale ones, are often underestimated. Besides, in the study by Pandolfi et al. (2014), higher risk of mortality was observed during reduced mixing layer height, caused by the

accumulation of anthropogenic primary air pollutants. In order to accurately investigate aerosol pollution processes in smaller areas with complex relief, high resolution quality measurements are needed. Available methods (e.g. remote sensing instruments such as sodars and ceilometers, aerosol lidars, aircraft measurements, ground-based scanning winds) for measuring

horizontal and vertical inhomogeneities in a complex relief are often expensive (Baasandorj et al., 2017) and do not provide all the necessary information if operated alone (there is a need for 3D data, continuous measurements and measurements of different aerosol properties) (Gohm et al., 2009; Wang et al. 2019). In order to acquire highly resolved spatial and temporal data, mobile measurements have been increasingly used in air quality monitoring, predominantly in urban areas (e.g. Alas et al., 2019a, 2019b, 2020 and the reference therein), yet rarely in hilly/mountainous rural regions (for instance Weimer et al.,

75 2009).

While much attention is given to urbanized deep valleys, only sparse data are available on people's exposure to air pollution in the countryside. According to the population density of Europe (Eurostat, 2011) which includes all EU's countries, the United Kingdom and four European Free Trade Association (EFTA) countries roughly one third (27 %) of the population are living in rural areas. More recent data on population distribution by degree of urbanisation from 2015 (DEGURBA, Eurostat,

2016) which includes EU member countries only, report very similar share of the population in rural areas (28 %) with a gradual increase by 1.7 % from 2010 to 2015 (Margaras, 2019).

The aim of this study is to quantify the influence of ground temperature inversions on spatiotemporal variability of wood combustion aerosol pollution in mountainous regions, an example of which is the model region Loški Potok, Slovenia. The study uses the Retje karst hollow as a representative example of a small-size terrain depression with a village situated at its

bottom. Besides the symmetrically shaped topography and smaller size of the hollow, allowing to study generally valid atmospheric and aerosol processes in relief depressions, the example is very convenient for studying the influence of residential wood combustion on local aerosol pollution, as the impact of other emission sources is expected to be rather small. The objectives of the study are: 1) to analyse wood burning aerosol pollution with quality high-resolution aerosol measurements, 2) to assess spatiotemporal variability of equivalent black carbon or eBC (term introduced by Petzold et al., 2013) and $PM_{2.5}$

during temperature inversion episodes and during unstable atmosphere, and 3) to determine the correlation between meteorological parameters with eBC and PM pollution.

To achieve these, we performed stationary and mobile measurements along the selected hollow. We identified temperature inversion events according to the vertical potential temperature gradients and determined mixing heights (MH) during these periods. Afterwards, correlation analysis between meteorological parameters and pollutants levels at the fixed stations and

along the hollow with respect to time of day was performed. Subsequently, the obtained results are discussed and finally, we provide conclusions and suggestions for rural relief depressions with limited self-cleaning air capacities.

## 2 Methods

### 2.1 Measurement site

The area selected for the study is a small, covering approx. 1.5 km², and shallow, less than 150 m deep, karst depression with a topography favourable for the formation of ground temperature inversions or cold air pools (CAPs), especially in winter. Terrain configuration of the karst depression in northeast direction with marked location of the Retje village is shown in Fig. 1. (c).

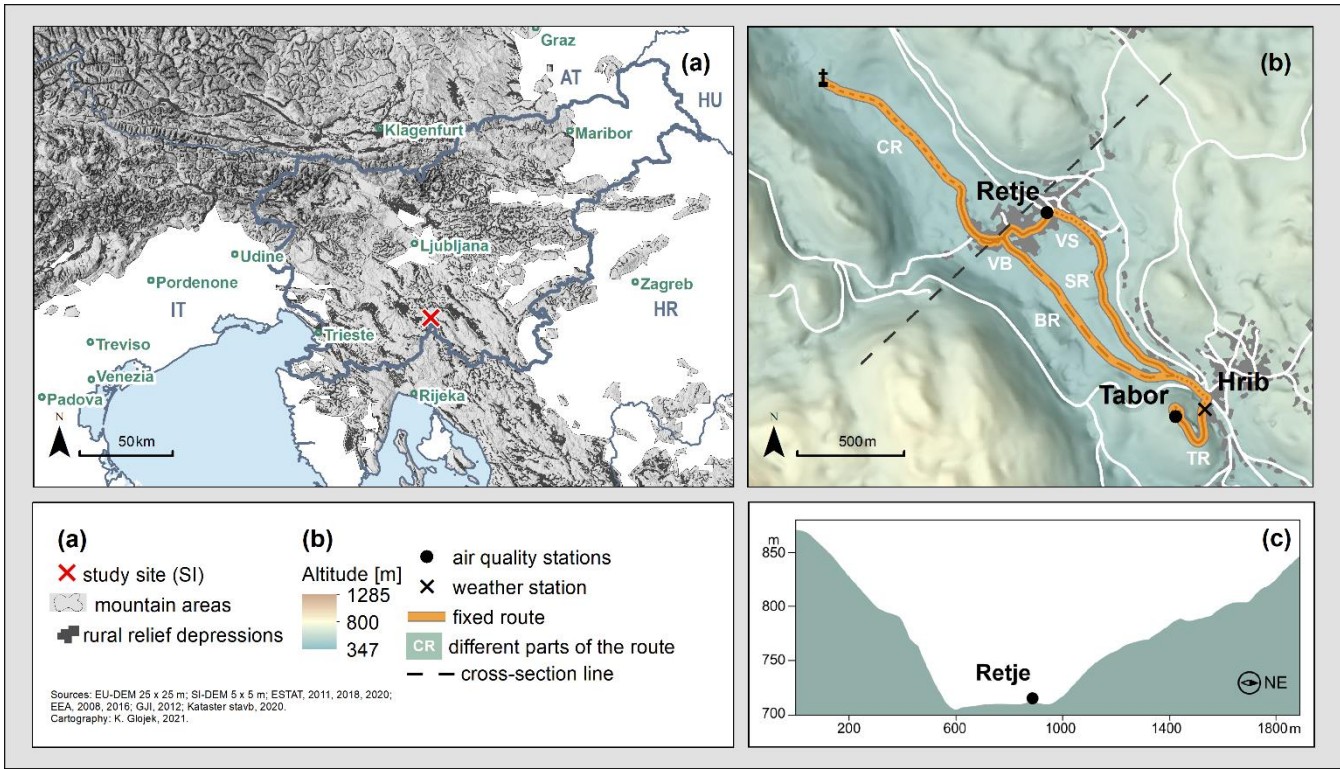

**Figure 1: (a) Location of the study site in Europe, (b) Topographic map of the studied area with the marked fixed route and stations, (c) Profile of the terrain across the Retje karst depression with marked village at its bottom.**

Surrounded by forests in all directions and therefore influenced almost solely by 243 households with 690 residents (SURS, 2018), the site is ideal to study impacts of residential wood burning on local air quality. Due to these characteristics,

measurements at Loški Potok can be taken as representative of the atmosphere found in rural hilly regions, where wood is used as a heating fuel. For more information about the measurement site the reader is referred to Glojek et al. (2018) and Glojek et al. (2020). In order to analyse the impact of ground temperature inversions on aerosol pollution, two temporary air quality stations with reference instruments were set up in the study area. One was placed at the bottom of the karst depression in the Retje village (45°42' 34.8''N 14°34'53.8''E, el. 715 m a.s.l.) as a rural village site and one on top of Tabor hill (45°42'01.5"N

14°35'19.7"E, el. 815 m a.s.l) for rural background atmosphere investigation. The Tabor site was selected for a rural background location due to its position, presumably above the ground temperature inversions; with minimal number of possible

emission sources in its immediate vicinity (cemetery, rectory building and one residential house); with limited accessibility to vehicles, yet reachable by mobile measurements; and due to possibility to place there the air quality station. Besides two air quality stations deployed in the study area, the 6 km long fixed route for mobile measurements was designed along the entire

hollow (Fig. 1. (b)). The route starts in the capital of the Loški Potok municipality i.e. in the settlement Hrib at south-eastern part of the karst hollow. The agglomeration is located along the regional thoroughfare and involves all the public buildings in the area (school, kindergarten, rest home, grocery store, post office, town hall and health centre). From there the fixed mobile route leads to the top of the hill Tabor passing the weather station in its immediate vicinity. After returning from the rural background air quality station Tabor, route continues in the direction of the village Retje (BR & VB). Leaving the lowest and

the widest part of the hollow, the route proceeds towards the chapel of St. Florjan at the north-western most point of the hollow. This part of the route (CR) is inaccessible to vehicles, and it is dominated by permanent pastures including some arable land. From there route returns to the residential area of the village Retje, which includes the bottom part of the hollow (BV) and the slopes, facing south (VS). The mobile measurements route covers the lower part of the slopes. The fixed route around the hollow ends in the settlement Hrib.

**2.2 Aerosol measurements**

In this study, the focus is on eBC mass concentrations and size-resolved measurements of particle number (PNC) and mass concentrations (PMC) of fine particle range in winter 2017–18. At the Retje air quality station, additional $PM_{10}$ mass concentrations are also available. In December 2017 and in January 2018 mobile measurements were carried out as well. The instrumentation used for the fixed stations and mobile platform are summarised in Table 1.

**Table 1: Description of reference and mobile instruments for eBC and PM measurements. Time is reported as local time (LT). Additional details in Glojek et. al., 2019; Alas et al., 2020 and in the Supplement.**

| Parameter | Instrument | Specifications | Time resolution | Measurement |
|---|---|---|---|---|
| **Equivalent black carbon (eBC)** | microAethalometer AE51, AethLabs | λ: 880 nm<br>σ: 12.1 $m^2 g^{-1}$ | 1 s | Mobile |
| | Aethalometer AE33, Magee Scientific | λ: 880 nm<br>σ: 7.7 $m^2 g^{-1}$<br>Inlet size cutoff: $PM_{2.5}$<br>$C_{M8060}$: 1.39 | 1 min | Stationary (reference, Retje and Tabor) |
| **Particulate matter (PM)** | Optical particle size spectrometer OPSS, 3330, TSI | Size range: 0.3–10 μm | 10 s | Mobile |
| | Mobility particle size spectrometer MPSS, TROPOS Ref. No. 1 & TSI 3785 | Size range: 0.01–0.8 μm & 0.01–0.6 μm<br>Inlet size cutoff: $PM_{2.5}$ | 5 min | Stationary (reference, Retje and Tabor) |

| High volume sampler, Digitel DHA-80 | $PM_{10}$ | 12 h (6:00–18:00 LT) | Stationary (reference, Retje) |

$\lambda$: wavelength $\qquad$ $C_{M8060}$: the correction parameter for the multiple scattering enhancement of the filter tape M8060
$\sigma$: mass absorption cross-section

### 2.2.1 Aerosol measurements at fixed stations

For both measurement stations, the inlet systems were equipped with a $PM_{2.5}$ inlet head which was connected to a 1.5 m Nafion®Permapure dryer (Wiedensohler et al., 2012) to maintain the relative humidity of the sampled air below 40 %. For eBC measurements, the Aethalometer model AE33 (Magee Scientific / Aerosol d.o.o.) was used. The instrument calculates the absorption and corresponding eBC mass concentrations from measurements of light transmission at 7 different wavelengths (370–950 nm). It uses a "dual-spot" technique to compensate the loading effect artifact in real-time (Drinovec et al., 2015). The flow rates of Aethalometers at the stations were set to 5 L $min^{-1}$ and the measurement time resolution to 1 min. The filter tape used was TFE-coated glass fiber filter (no. M8060). The mass absorption cross section ($\sigma$) of 7.77 $m^2$ $g^{-1}$ was used to convert the optical measurement at 880 nm to eBC mass concentration. The Aethalometer AE33 also served as reference for eBC mass concentration for the mobile platform.

Parameters of PNC and their size distributions (PNSD) were measured using the Mobility Particle Size Spectrometers (MPSSs) following the recommendations given in Wiedensohler et al. (2012). The quality assurance of the instruments and measurements were done according to Wiedensohler et al. (2018). The MPSS classifies electrical particle mobility typically in the range 0.01–0.80 µm. Using the standardized bipolar charge distribution, the PNSD can be calculated (Wiedensohler, 1988). A reference MPSS designed by Leibniz Institute for Tropospheric Research (TROPOS Ref. No. 1, Hauke medium Differential Mobility Particle Sizer (DMA)) equipped with a butanol-based condensation particle counter (CPC, TSI model 3772) and a Total CPC (TSI model 3010) were set up at the Retje rural village station, whereas at the Tabor station a TSI MPSS (DMA model 3081) with water-based CPC (TSI model 3785) was deployed. The TROPOS MPSS Ref. No. 1 uses a $^{85}$Kr neutraliser in bipolar diffusion charger, while the TSI MPSS at the Tabor station had an x-ray source to charge aerosol particles before they enter the DMA. The TROPOS-type MPSS uses high-voltage supply of positive polarity, whereas TSI MPSS has a negative polarity. The sheath flow rate of 5 L $min^{-1}$ and an aerosol flow of 1 L $min^{-1}$ were set up for TROPOS MPSS with particle size range 0.01–0.80 µm. At Tabor station the sheath air to aerosol flow ratio was set to 4.1:1 l $min^{-1}$, resulting in particle size range 0.01–0.60 µm. Both instruments were operated with a time resolution of 5 min. The data from both stations were processed with the TROPOS software with a linear multiple charge inversion algorithm (Pfeifer et al., 2014) and additional corrections for internal and sampling losses due to diffusion and sedimentation as described in Wiedensohler et al. (2012). In case of the TSI MPSS at Tabor station the inversion for an x-ray matrix was included as well (Tigges et al, 2015). The quality of the MPSS data was assured by laboratory inter-comparisons with reference instruments performed by the World Calibration Center for Aerosol Physics (WCCAP) at TROPOS in Leipzig, Germany (Fig. S2 to S6 in the Supplement). From the PNSD measured at the Retje site $PM_1$ mass concentrations were calculated. The results of the procedure described in Sect.

2.2.2 served as PM reference for the intercomparisons with the mobile instruments. More information about the aerosol measurements at the fixed stations can be found in Glojek et al., 2019, 2020 and in the Supplement.

Additional $PM_{10}$ measurements were obtained according to EN 12341 standard using a high-volume sampler (Digitel, DHA-80). The PM particles were collected on preheated quartz fiber filters (150 mm diameter) at a flow rate of 500 l h$^{-1}$. 12-hour sampling was performed, from 6:00 to 18:00 and from 18:00 to 6:00 local time (LT). The filters were stored frozen until analysis.

## 2.2.2 Mobile eBC and $PM_{2.5}$ measurements

The mobile measurements were done using the TROPOS aerosol backpack equipped with portable instruments (Alas et al., 2019a, 2019b; Alas et al., 2020). On top of the backpack, the aerosol is sampled with 1 m long stainless steel inlet. Inside the backpack, eBC mass concentration is measured with microAethalometer AE51 (AethLab). The portable absorption photometer measures the attenuation of light at wavelength of 880 nm through a particle-loaded T60 Teflon coated glass fiber filter and converts it to eBC mass concentrations using $\sigma$ of 12.5 m$^2$ g$^{-1}$ (provided by the manufacturer). Before entering the AE51 the aerosol sample passes through a silica-gel drier which dries the aerosols and dampens the effect of sudden changes in relative humidity and temperature (Cai et al., 2014; Düsing et al., 2019), which could produce false signals in the instrument. MicroAethalometer was operated with a flow of 150 ml min$^{-1}$ and time stamp of 1 s. In order to minimize the impacts of single events that do not represent the typical concentration profile of the area and due to high noise of eBC mass concentrations at 1 s resolution, the data were processed to 10 s medians.

To obtain PM mass concentrations, optical particle size spectrometer (TSI OPSS, model 3330) was used. The instrument is connected to another aerosol line inside the backpack, which is not dried. It is an optical instrument that measures the optical particle diameter according to the light scattered and related refractive index of the calibration aerosol, typically polystyrene latex (PSL) spheres with refractive index (ñ) of 1.59. However, the optical properties of ambient aerosols largely differ from the PSL, which results in wrong sizing in terms of the volume equivalent diameter. Furthermore, the OPSS measures an PSL equivalent size range of 0.3–10 μm, while the accumulation mode is only partially measured, and the Aitken mode is not measured at all. This means that PM calculation from the OPSS PNSDs generally lead to an underestimation of the mass concentrations, which can be signifcant. To address this issue, the OPSS data were corrected according to the procedure briefly described in the latter part of this section, which follows the methodology presented in Alas et al., 2019a. During the Loški Potok campaign, the OPSS had a flow of 1 l min$^{-1}$ and 10 s time resolution. The position of the mobile measurements was obtained by a GPS device placed on top of the inlet. All instruments together with the GPS unit were synchronized and controlled by a microcomputer where the data was also logged.

Simultaneous mobile measurements with two instrumented backpacks were carried out 3 times a day along a fixed route in the karst depression Retje. The run in the morning lasted from 6:30 to 9:00, at noon from 12:00 to 14:00 and in the evening from 17:00 to 19:00 LT. To obtain representative information, daily measurements along the depression were conducted 107 times, totalling to 642 km. Results of the convergence analysis (see Alas et al., 2020 and the reference therein for the assessment

process) for all data points in the Retje village and at Tabor hill are depicted in Fig. S12. The number of runs when the cumulative median concentrations of randomly selected runs deviate less than 50 % from the median of all the runs at the given location was selected for a threshold. According to visual analysis, the data converged faster for the hill Tabor compared to the village Retje. Thus, the minimum number of runs required to obtain a representative image of the air quality on the Tabor hill is 46 for eBC and 52 for $PM_{2.5}$, whereas for the Retje village it is 65 for eBC and 58 for $PM_{2.5}$.

**Table 2: Air quality measurement stations with the "reference" aerosol instruments at the study site.**

| Intercomparison location | Station type | Elevation (m) | Instrument | Duration of intercomparison (min) |
|---|---|---|---|---|
| Retje | Rural village | 705 | AE33 Aethalometer | 20 |
|  |  |  | TROPOS-type MPSS with TSI CPC & TCPC 3772 |  |
|  |  |  | Digitel DHA-80 |  |
| Tabor | Rural background | 815 | AE33 Aethalometer | 10 |
|  |  |  | TSI MPSS with TSI CPC 3785 |  |

The route (see Fig. 1. (b)) led past fixed stations with the following reference instruments, the AE33 Aethalometer and the MPSS (Table 2). During each run the carriers of the backpacks intercompared the mobile AE51s with the AE33 Aethalometers and the OPSSs 3330 with the MPSSs. Intercomparison at the Retje site (the rural village station) lasted 20 min and at the Tabor site (the rural background site) 10 min. The methodology used to assure high-quality mobile measurements of eBC and PM mass concentrations is provided in Alas et al.(2019a).

In order to obtain accurate eBC mass concentrations, mobile filter photometers were corrected for the filter-loading effect (FLE). The correction algorithm presented by Virkkula et al. (2007) was used, with a loading parameter (k) of 0.005 for the AE51 based on Drinovec et al. (2017). Besides diesel dominated aerosols, the selected value of k is representative for freshly emitted particles from wood burning as well. Detection of the filter-loading effect, correction, and correlation between the microAethalometers AE51 against reference instruments (the AE33 Aethalometers) used in the Loški Potok campaign are published in Alas et al. (2020).

As previously stated, $PM_{2.5}$ mass concentrations were derived from the OPSS measurements that were corrected by the aerosol type-dependent complex refractive index (ñ) and unique fine-mode volume correction factors ($CF_{f, vol}$). The real part ($ñ_{re}$) of the OPSS ñ was adjusted to 1.52 and the imaginary part ($ñ_{im}$) to i0.02 for all size bins. Both values are typical for rural, residential areas containing absorbing aerosols (Ebert et al., 2004; Lu et al. 2015). However, applying one corrected ñ for particles in range 0.3–0.8 µm, as well as for the particles larger than 0.8 µm, could lead to high uncertainties due to unknown particle shape and light absorbing particle composition in the supermicrometer range. After the ñ correction had been applied on the OPSS PNSD, using the Mie Theory, the optical diameters were converted to geometric mean volume equivalent diameters for the submicrometer size range (the correction of the data is illustrated in the Supplement Fig. S7, S8 and S9). The $CF_{f, vol}$ corrects the OPSS constraint of measuring only a fraction of the fine mode particles (from 0.3 to 10 µm). Therefore, $CFs_{f, vol}$ were calculated as a ratio between the fine mode of the OPSS (from 0.4 to 0.8 µm) with the fine mode of MPSS (from

230 0.001 to 0.8 μm) during each intercomparison period, since the variability of PVSD requires the use of unique $CF_{f, vol}$ for each individual run (Alas et al., 2019a). As there are no significant differences in aerosol sources along the hollow (see Glojek et al., 2018) and due to instrument failures at the second measurement location, data points of the whole selected route were corrected with the correction factors derived at the Retje rural village station. The relevance of the use of correction factors obtained at a single measurement site (Retje) for the entire route was confirmed by the comparison of the particle volume size

distributions (PVSDs) measured by the MPSS at the Tabor rural background station. Namely, PVSDs at both measurement sites peaked around 0.3 μm (see Supplement Fig. S8). Once all the corrections were applied for each run, $PM_1$ mass concentrations were calculated. Since the organic aerosols prevail in the studied area, a particle density of 1.5 g $cm^{-3}$ (Turpin and Lim, 2001) was used for the calculation. The absolute value of $PM_1$ was added to the absolute value of $PM_{1-2.5}$ to obtain $PM_{2.5}$ mass concentrations. Corrected $PM_1$ mass concentrations of the OPSS are comparable to the reference MPSS.

Correlation between median $PM_1$ mass concentrations obtained from the OPSS and MPSS during each intercomparison time is high, with $R^2 > 0.8$ and slope of 1.12. Additionally, $PM_1$ mass concentrations derived from the MPSS measurements were compared to the 12-hour $PM_{10}$ data, measured by the High-Volume Sampler Digitel DHA-80. These two parameters correlate well too, with $R^2=1$ and slope of 0.9 (Supplement Fig. S7). The relationship shows that most of the mass concentration of $PM_{10}$ is already found in $PM_1$. Despite considerably improved calculated $PM_{2.5}$ mass concentration values of the OPSS after

applied corrections, some uncertainties remain, primarily due to the particles non-sphericity, assumed particle density and the use of the same time-dependent correction factors for the whole route. With an increase in particle size, the shape of the particles becomes even more non-spherical and thus the refractive index correction based on the Mie theory, assuming spherical particles, is not possible. Due to this high uncertainty, $PM_{10}$ mass concentrations from the OPSS data were not calculated for this study.

**2.3 Meteorological parameters**

Hourly and daily averages of meteorological parameters recorded from air quality stations in Retje and at Tabor (ambient temperature, pressure, and relative humidity) and from the weather station of the non-governmental Society for Weather and Climate Research (Društvo za raziskovanje vremena in Podnebja, 45°42′4″ N 14°35′27″ E, el. 775 m a.s.l.) (ambient temperature, pressure, relative humidity, wind speed and direction and the sum of precipitation) were computed. The location

of the weather station is marked with the black cross (Hrib in Fig. 1. (b)). To determine temperature inversion episodes, mobile temperature measurements with a temperature sensor (DS1922L iButton, Maxim Integrated) were considered as well. Besides the observed meteorological variables, heating degree days (HDD) were determined from the daily mean temperatures at the stations. The index was calculated according to Eurostat (2019) as well as ARSO (2020) methodology. The former uses a daily threshold value, defined as the lowest daily mean air temperature not leading to indoor heating, of 15 °C and the latter 12 °C.

### 2.3.1 Detection and selection of temperature inversion periods

Temperature inversion periods in the selected karst depression were detected by analysing the calculated potential temperature gradient, obtained from ground-based observations of temperatures at different elevations, and the mobile temperature measurements along the hollow. The ratio of temperature difference between stations at different heights is a good indicator of the stability of the boundary layer during temperature inversions as shown in, for instance, Petkovšek, 1978; Whiteman et al., 1999, 2014; Pandolfi et al., 2013; Holmes et al., 2015; Largeron and Staquet, 2016. In our study potential temperature differences between the Retje station at the bottom of the karst depression, the Hrib station on the slope and the Tabor station on top of the hill were calculated ($\Delta\theta/\Delta Z$). Mobile measurements of potential temperatures in December along the Retje karst depression were used as supplementary information to determine atmospheric conditions during the runs. If potential temperatures during mobile measurements along the karst depression were increasing with altitude (positive gradient), the runs were classified as runs during the temperature inversions. However, if potential temperatures were decreasing (negative gradient), the runs were classified as runs with an unstable atmosphere. To conform to the defined atmospheric conditions other meteorological variables were considered as well. Number of classified runs during temperature inversion and during unstable atmosphere is represented in Table 3.

**Table 3: Number of runs in winter 2017–2018 classified as runs during temperature inversion and runs during unstable atmosphere. The number of runs with strong temperature inversion is shown in brackets.**

| Time of day (LT) | | Number of runs with temperature inversion | Number of runs with unstable atmosphere |
|---|---|---|---|
| Morning | 6:30–9:00 | 20 (13) | 15 |
| Afternoon | 12:00–14:00 | 7 (4) | 12 |
| Evening | 17:00–19:00 | 19 (10) | 16 |
| | **altogether** | **46 (27)** | **43** |

Out of the total 107 runs performed along the hollow, 98 were valid, i.e. without measurements failures (e.g. leak in the system, wrong GPS route tracking, full filter etc.), and classified according to the meteorological conditions. Forty-six runs were classified as runs during temperature inversion, from which 27 with strong temperature inversion (the numbers indicated in brackets). The criteria for further separation of the temperature inversions is determined by the potential temperature gradient above 1 K $(100\ m)^{-1}$ and by the change of the weather regime during the run. Thus, run with strong temperature inversion stands for the run with a potential temperature gradient above 1 K $(100\ m)^{-1}$ during the whole run with no weather changes or intensive mixing of atmosphere during the run. Forty-three runs had a negative potential temperature gradient during the whole run and were therefore classified as runs with unstable atmosphere. During 9 of the runs atmospheric conditions changed from stable to unstable or vice versa and thus were not taken into account in the analysis according to meteorological conditions.

### 2.3.2 Determination of the mixing height during temperature inversions

Vertical eBC profiles allow the determination of the height up to which the surface pollutants are well mixed (Ferrero et al., 2010 and the references therein). This atmospheric boundary layer height (ABL) is also called the mixing height (MH). It indicates pollutant accumulation in the planet boundary layer (PBL), which is the layer of the atmosphere directly influenced by the Earth's surface. The MH is related to meteorological parameters and surface roughness which are governing the behaviour of the PBL (Seibert et al., 2000). During the day with fair weather, the MH represents the height of the convective boundary layer (CBL) decoupled from the free atmosphere with a strong temperature inversion. With nocturnal radiative cooling, a statically stable boundary layer (SBL) forms at the bottom of the CBL. Above the SBL, a statically neutral layer from a previous day remains, called residual layer (RI) (Stull, 2017). Hereinafter referred MH follows the definition presented by Seibert et al. (2000) and Ferrero et al. (2011). It presents a layer affected by local emissions, regardless of the time of day. Its height may be lower than a depth of PBL, especially over complex terrain.

Previous studies (Kim et al., 2007; Angelini et al., 2009; Ferrero et al., 2010; 2011; Ferrero et al., 2016) demonstrated the accuracy of particle and eBC derived MH method, namely, atmospheric particles indicate the atmospheric dispersion state. Particles in the mixing layer accumulate due to a very stable atmosphere. Therefore, a significant difference in concentration levels between the surface boundary layer compared to the free atmosphere commonly occurs (Emeis et al., 2008; Summa et al., 2013). This sharp decrease in concentrations is defined as a MH (Balsley et al., 2006; Kim et al., 2007; Yang et al., 2017). Mobile measurements of eBC and potential temperature along the hollow were used for the determination of MHs during the selected temperature inversion episodes. MH was calculated by means of the gradient method from 5 m averaged eBC pseudo-vertical profiles (some details on spatial averaging are provided in Sect. 2.4). The profiles were named pseudo-vertical as they were obtained with mobile measurements along the slopes of the hollow and not with commonly used methods for vertical profile measurements, e.g. radio-soundings, balloons, UAVs or lidars.

The strongest gradient of the eBC concentration profile was chosen as the MH. To confirm the reliability of eBC-derived MH, mixing heights for the selected temperature inversion episodes in December were determined by mobile potential temperature measurements as well. As with the eBC-derived MH, MH estimated from potential temperature profiles (θ-MH) was determined as the height of sudden change in temperature (derivatives determined as MH are presented in the Supplement Table 1). The method limitations are described in more detail in Sect. 3.1.1 under Results.

### 2.4 Spatial and statistical analysis

Concerning the second goal, mobile measurements were spatially averaged according to the method described in Alas et. al., 2019a and Alas et al., 2019b. Firstly, a pre-determined route with 10 m equidistant points in QGIS 2.18.22 was created. These points were used as centre points for spatial aggregation performed in R programming language. Prior to spatial aggregation, mobile runs were separated into two groups according to atmospheric conditions: runs occurring during temperature inversions and runs with an unstable atmosphere (for the classification of runs with respect to meteorological conditions see section 3.3.1).

Subsequently, the median of all data points in a radius of 10 m around each centre point was calculated. Results are moving circular medians of eBC an $PM_{2.5}$ mass concentrations along the route calculated separately for all the runs during strong temperature inversions (27) and for all the runs during unstable atmosphere (43). In addition, average ambient ratios of eBC to $PM_{2.5}$ for the whole hollow were calculated.

For the goal to establish association between aerosol pollution and meteorology, correlation analyses between meteorological variables with eBC and PM mass concentrations for the whole measurement period and for the selected days with temperature inversion (temperature inversion days) and days with unstable atmosphere (non-inversion days) was performed. An ordinary least squares (OLS) linear regression was applied using Python programming language with the "scipy" module.

Finally, we investigated the number of people living in rural hilly and mountainous areas across Europe. An enquiry analysis of European Digital Elevation Model of 25 x 25 m (EEA, 2007) was performed. The analysis was conducted for the European mountain areas as defined by European Environmental Agency (2008) excluding Turkish peninsula and islands except those that are part of the mountain range (e.g. Sicily), and Iceland. Relief depressions were identified with the TPI-based Landform tool in the software environment SAGA GIS 2.3.2. Landforms classes such as drainages, valleys and plains bigger than 0.4 $km^2$ were selected for further analysis. As we were interested in populated rural relief hollows uninhabited and urban areas were eliminated with the help of population density grid of 1 $km^2$ (Eurostat, 2011) and with the spatial classification Degree of urbanisation (DEGURBA, Eurostat, 2016). Data manipulation was done using Python and ArcGIS 10.8.1. The results are shown in the Supplement (Fig. S15).

## 3 Results

### 3.1 Meteorological conditions, PM₁₀ and eBC mass concentrations

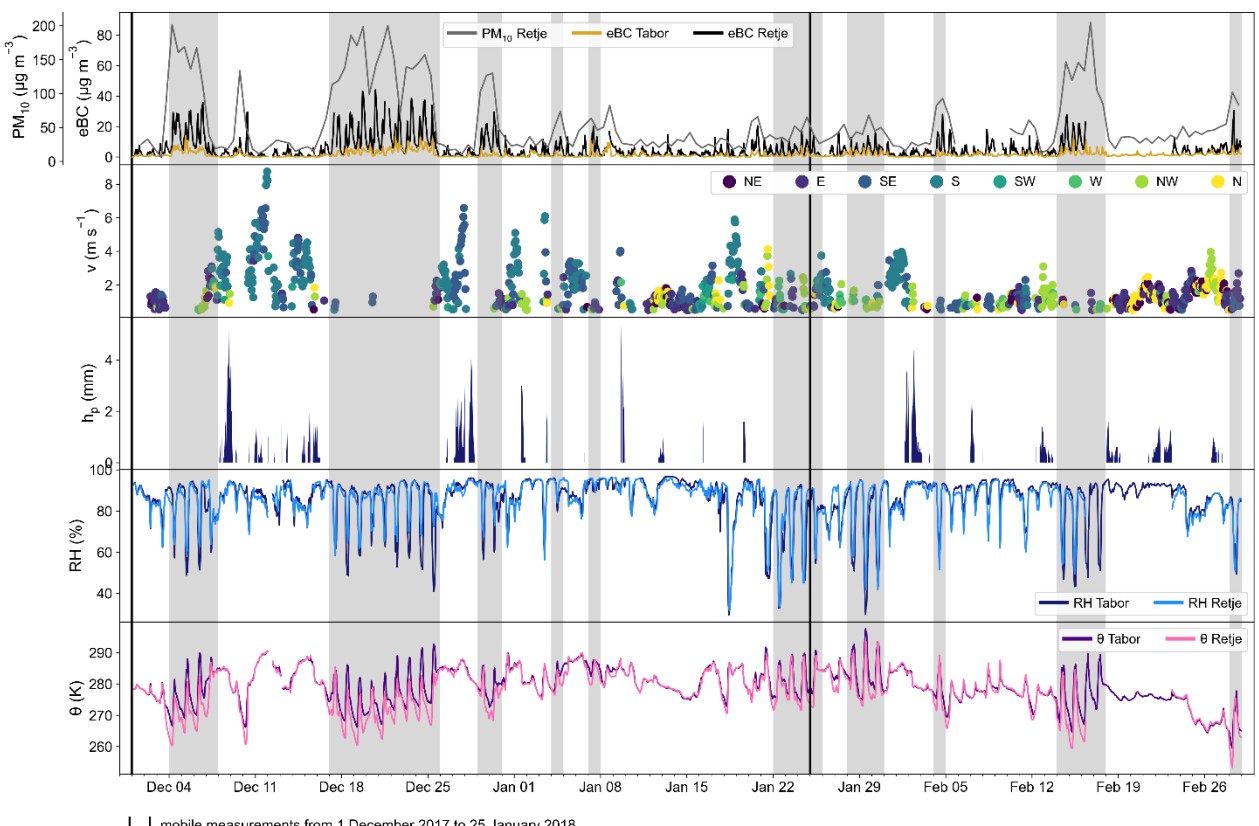

**Figure 2: Time series of meteorological and air quality data for winter period (December 1, 2017–March 1, 2018): the 12-hour PM₁₀ mass concentrations at Retje, the hourly mean eBC mass concentrations at Retje and Tabor, wind speed (v) and direction at Hrib, precipitation (hₚ) at Hrib, relative humidity (RH) and potential temperature (θ) at Retje and Tabor. The shaded areas represent temperature inversion periods.**

Twelve-hour PM₁₀ and hourly averages of eBC mass concentrations and meteorological conditions during winter (1 December 2017–1 March 2018) and during the mobile measurement period (1 December 2017–25 January 2018) are illustrated in Fig. 2. Shaded areas represent selected temperature inversion periods, when the potential temperature at the station Retje at 715 m a.s.l. (pink line) was lower than the potential temperature at the top of the hill Tabor at 815 m a.s.l (purple line). As described in the method section (2. 3.) and in Glojek et al. (2018) wind data were collected at Hrib at 775 m a.s.l. The location of the wind measurements out of the hollow is not the most representative of the wind characteristics in the very basin. Thus, wind speeds in the hollow are expected to be lower due to topography, which prevents effective ventilation.

Rapidly changing meteorological conditions and pollutants mass concentrations were observed during the whole winter period 2017–2018. eBC mass concentrations at the bottom of the hollow in the village Retje (black line) were consistently higher

than at the rural background station on top of the hill Tabor (yellow line). It was at Tabor station where a smaller range in eBC mass concentration levels was recorded overtime as well. Mean hourly eBC concentrations at Tabor were ranging from few hundred ng m$^{-3}$ to 20 μg m$^{-3}$, while eBC concentrations at Retje were in the range from 0.9 μg m$^{-3}$ (25$^{th}$ percentile) to above 40

μg m$^{-3}$. PM$_{10}$ levels varied between below 10 and up to 205 μg m$^{-3}$ at Retje village (grey line).

Intensity of sources and meteorological variability are followed by changes in pollutant concentrations. The highest eBC mass concentrations at both measurement sites were observed during two longer periods of stable atmosphere in December 2017. Similar results were found for PM$_{10}$ as well. There was also a third distinct increase in pollution levels at Retje during the longest temperature inversion period in February. However, the eBC data for this period are partly missing. The first period

lasted from 4 December to 8 December, the second period from 17 December to 25 December and the third period from 14 February to 18 February. The second period in December presents the longest period with temperature inversion of the whole campaign. All of these periods were characterized by an anticyclonic weather with none or weak, less than 0.7 m s$^{-1}$ wind speeds, the absence of precipitation, and low temperatures at the bottom of the hollow, reaching a trough of –18.4 °C on February 15 at the Retje village station. Due to lower temperatures at the bottom of the hollow than on top of the hill, the

relative humidity in the hollow was higher than on Tabor hill. Morning and evening temperatures in the hollow dropped regularly below the dew point during temperature inversions, forming radiation fog. Regular diurnal eBC, temperature and relative humidity oscillations could be seen during temperature inversions. The valley floor was covered with snow during all three temperature inversion episodes. The period between the first and the second longer temperature inversion in December i.e., from 8 December to 16 December, was characterized by an intense frontal activity with strong, from 2 m s$^{-1}$ (25$^{th}$ percentile)

to 4 m s$^{-1}$ (75$^{th}$ percentile) south to southeast winds with gusts above 9 m s$^{-1}$ and high snow precipitation with 61 cm of new snow recorded. After 10 December there was a strong thaw with warm air masses. After the second temperature inversion period in December the temperatures rose again and stayed above the long-term average for the whole month of January and at the beginning of February (see Glojek et al., 2018). Precipitations in the form of rain were above the long-term average in comparison with the usual amount of rainfall in January (Cegnar, T., Knez, 2018). Out of 14 precipitation days in January,

only 4 days had snowed. In January and at the beginning of February periods with temperature inversions were shorter compared to December, lasting no longer than 4 days, with higher temperatures and lower relative humidity, respectively, 2.5 °C higher and 7.6 % lower mean values at all the stations. Moreover, the PM$_{10}$ and eBC mass concentration levels during January and February temperature inversions were lower compared to temperature inversions in December, except from 14 February to 18 February when PM$_{10}$ mass concentrations reached maximum levels. In Retje, mean PM$_{10}$ and eBC levels were,

on average, more than 2 times lower, with mean of 133.5 ± 42.3 μg m$^{-3}$ and 12.9 ± 10.7 μg m$^{-3}$, respectively, for December inversions and mean of 56.2 ± 19.9 of PM$_{10}$ and 6.3 ± 6.7 μg m$^{-3}$ of eBC for inversions in January and February. Due to missing data, the longest temperature inversion episode in February was not included in the calculation. At Tabor the difference between mean value of 3.6 ± 3.3 μg m$^{-3}$ for December inversions and mean of 2.3 ± 2.7 μg m$^{-3}$ for inversions in January and February was 1.3 μg m$^{-3}$ of eBC. In February temperatures dropped, especially in the last days of the month when temperature

in Retje dropped below -24.7 °C, reaching minimum temperatures for the winter of 2017–2018. Cold and moist east and

northeast winds prevailed over the area bringing snow precipitation respectively. In February there were more than 19 days with precipitation, including 17 days of snowfall, and 26 days of snow cover. Cloudy weather prevailed during the whole month. Temperature inversions occurred mainly within the first and the second third of February.

### 3.2 Spatiotemporal variability of eBC and PM$_{2.5}$ during temperature inversion and during unstable atmosphere

As already seen in time series of eBC and PM$_{10}$ mass concentrations at the stations in Fig. 2, concentration levels vary in time as the conditions in atmosphere change. The highest eBC mass concentrations were observed during temperature inversions while the lowest values were detected during unstable atmosphere with the negative vertical temperature gradient. Fig. 3 (upper part) shows the spatial distribution of eBC and Fig. 3 (lower part) illustrates PM$_{2.5}$ mass concentrations during different exemplary meteorological conditions along the hollow. The spatial average of pollutants levels for all selected runs is

presented.

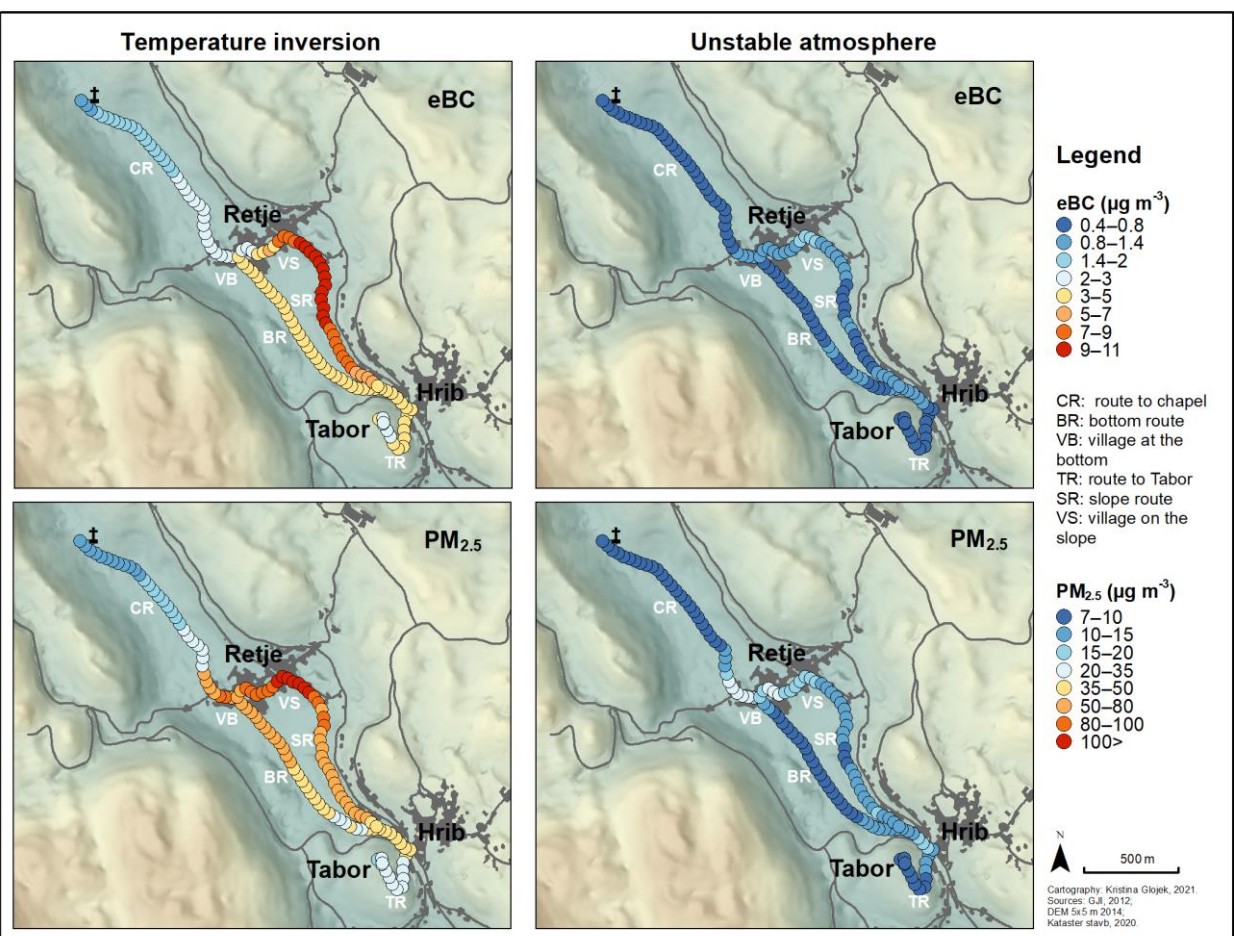

**Figure 3: Spatial distribution of eBC (upper part) and PM$_{2.5}$ (lower part) mass concentrations during temperature inversions (left side) and during unstable atmosphere (right side) along the Retje karst hollow. The spatial average of pollutants levels for all selected runs is presented.**

During winter temperature inversions eBC and $PM_{2.5}$ mass concentration levels build up in the whole hollow with the mean value of $4.5 \pm 2.6$ µg m$^{-3}$ of eBC and $48.0 \pm 27.7$ µg m$^{-3}$ of $PM_{2.5}$. However, considerable differences are observed among different positions in the relief depression. The highest pollutant mass concentrations are observed at south-eastern end of the hollow, in the village Retje, near the air quality station (VS) and along the first part of the slope road (SR) that connects the village Retje with the settlement Hrib. The $PM_{2.5}$ mass concentrations are the highest close to the agglomeration Retje (Fig. 3. on the lower left-hand side), whereas the eBC mass concentrations are the highest along the slope road (Fig. 3. on the upper left-hand side). Mass concentrations in this area reach 11 µg m$^{-3}$ of eBC and 140.3 µg m$^{-3}$ of $PM_{2.5}$ on average. Contrary to pollutants maxima, the lowest eBC and $PM_{2.5}$ mass concentrations were measured out of the village at the chapel Florjan in the north-western-most part of the hollow. Mean eBC mass concentrations of 1.3 µg m$^{-3}$ and mean $PM_{2.5}$ of 10.9 µg m$^{-3}$ were 9 to 13 times lower than at the village Retje. On Tabor hill at the other, south-easterly end of the hollow above the settlement Hrib, mean values of eBC and $PM_{2.5}$ were 2 µg m$^{-3}$ and 17 µg m$^{-3}$, respectively. Mean eBC mass concentrations of 4 µg m$^{-3}$ were observed along the route to the village Retje (BR). Levels of $PM_{2.5}$ were increasing from 32 to 60 µg m$^{-3}$ along the route towards the village (BR), which lies at a lower altitude. The eBC trend, however, depends on the time of the day (Fig. 4.). In the morning, from 6:30 to 9:00, eBC levels with the mean value of 3 µg m$^{-3}$ remain constant along the whole bottom route (BR), whereas around noon and in the early evening, eBC mass concentrations change along the route. As seen with the $PM_{2.5}$ levels, early evening eBC mass concentrations increase from 6.6 µg m$^{-3}$ to 11.6 µg m$^{-3}$ in the direction of the village Retje as well. Conversely, around noon eBC mass concentrations are higher on the part of the route leading to the settlement Hrib (SR), which is at a higher altitude, increasing from 4.0 to 7.5 µg m$^{-3}$.

**Table 4:** Summary of eBC and PM$_{2.5}$ mass concentrations measured along the route of the Retje karst depression. Mean with standard deviation (AM ± STD), median (MED) and average eBC/PM$_{2.5}$ ratio of the whole route during temperature inversions and during unstable atmosphere shown for all the runs (all) and separately according to different run times. MIN and MAX represent value and location of pollutants minimum and maximum along the route.

| Measurement period | | eBC (µg m$^{-3}$) | | | | PM$_{2.5}$ (µg m$^{-3}$) | | | | eBC/ PM$_{2.5}$ |
|---|---|---|---|---|---|---|---|---|---|---|
| | | mean ± std | med | min | max | mean ± std | med | min | max | |
| Temperature inversion | all | 4.5±2.6 | 4 | 1.3 (C) | 11 (VS) | 48.0 ± 27.7 | 43.7 | 10.9 (C) | 140.3 (VS) | 0.094 |
| | morning | 3.4 ± 2.8 | 2.8 | 1.1 (C) | 10 (SR) | 41.8 ± 25.1 | 36.5 | 7.7 (C) | 120.5 (VS) | 0.089 |
| | afternoon | 4.9 ± 2.2 | 4.4 | 1.8 (CR) | 12 (VS) | 49.7 ± 26.8 | 43.8 | 15.3 (C) | 182.7 (VS) | 0.085 |
| | evening | 9.3 ± 4.8 | 8.3 | 1.7 (TR) | 22 (VS) | 165.2 ± 156.7 | 113.7 | 12.3 (T) | 560.7 (VB) | 0.080 |
| Unstable atmosphere | all | 0.9 ± 0.3 | 0.8 | 0.4 (T) | 1.8 (VS) | 11.9 ± 4.2 | 10.9 | 7.7 (TR) | 29.2 (VB) | 0.076 |
| | morning | 0.5 ± 0.3 | 0.5 | 0.2 (C) | 1.9 (VS) | 11.2 ± 3.3 | 10.6 | 5.7 (T) | 23.8 (VB) | 0.071 |
| | afternoon | 0.7 ± 0.3 | 0.6 | 0.3 (C) | 1.7 (VS) | 10.9 ± 5.2 | 9.0 | 5.0 (C) | 36.1 (VB) | 0.067 |
| | evening | 1.3 ± 0.3 | 1.2 | 0.8 (C) | 2.1 (VS) | 15.3 ± 6.2 | 14.2 | 7.7 (BR) | 39.3 (VB) | 0.062 |

C & CR: chapel & route to chapel   BR: bottom route   VB: village at the bottom
T & TR: Tabor & route to Tabor   SR: slope route   VS: village on the slope

During unstable atmosphere (right side of Fig. 3.) mean mass concentrations of eBC and PM$_{2.5}$ in the entire hollow do not reach 1 µg m$^{-3}$ of eBC and 12 µg m$^{-3}$ of PM$_{2.5}$. Nonetheless, increased eBC and PM$_{2.5}$ mass concentrations in the village Retje are observed, reaching 1.8 µg m$^{-3}$ of eBC and 29.2 µg m$^{-3}$ of PM$_{2.5}$ on average. A slight increase in mass concentrations is seen near the settlement Hrib as well, with mean eBC of 0.8 µg m$^{-3}$ and mean PM$_{2.5}$ of 15 µg m$^{-3}$. The ratios of eBC/PM$_{2.5}$ during the winter 2017–2018 campaign observed in the model region Loški Potok ranged from 0.062 to 0.094. Relationship between eBC and PM$_{2.5}$ mass concentrations per coded locations along the route is shown in Fig S11. Correlation is good for all locations ($R^2 > 0.6$ or $R^2 = 0.6$), especially for non-traffic ones ($R^2 > 85$). The eBC mass fraction in PM$_{2.5}$ is higher at Tabor (T & TR) and Hrib (H) compared to locations in the hollow.

A deeper analysis of the pollutants' spatial distributions during temperature inversions is shown in Fig.4. The determined line segments along the fixed route inside the Retje hollow are listed and described in Table 4 with their spatial locations marked in Fig. 5. Here, the goal was to show horizontal variability of eBC and PM$_{2.5}$ along the hollow with respect to different times of the day. To this end, line segments at similar relative heights were selected (a maximum difference of 30 m per route segment; see Table 5).

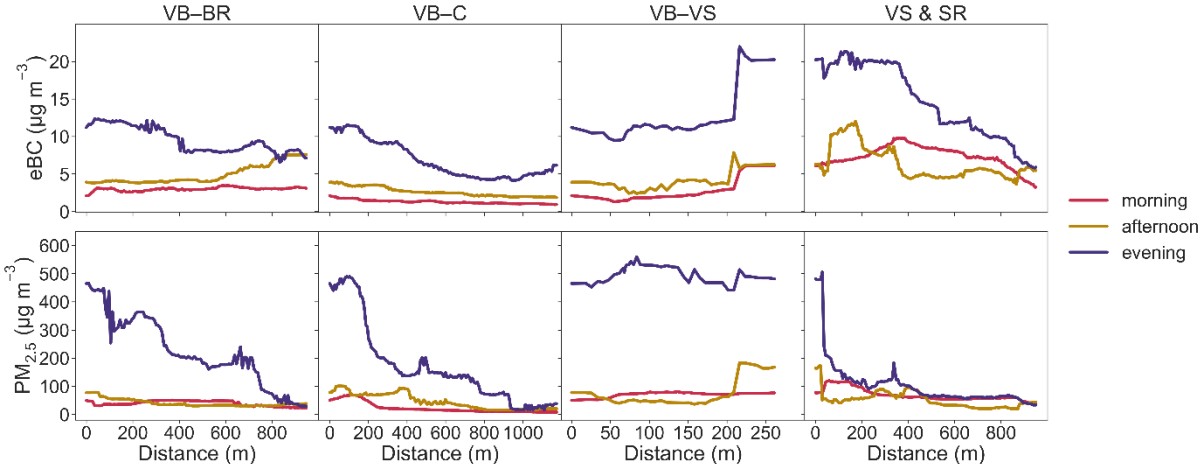

**Figure 4: Variability of eBC (top panels) and PM$_{2.5}$ (bottom panels) mass concentrations during morning (6:30–8:30 LT, red line), afternoon (12.00:14.00 LT, yellow line) and evening (17:00–19:00 LT, purple line) winter-time temperature inversion along different parts of the fixed route in the Retje hollow.**

**Table 5: List and description of the different route segments along the Retje hollow.**

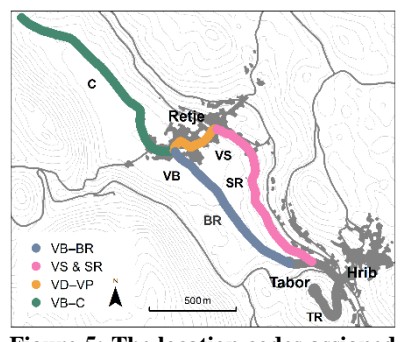

| Location code | Descriptions | Altitudes (m) | eBC/PM$_{2.5}$ |
|---|---|---|---|
| **VB–BR** | Bottom route in a village Retje-Hrib direction. | 705.9–741.5 | 0.12 |
| **VB–C** | Line segment in a village Retje-chapel st. Florjan direction. | 706.7–728.9 | 0.11 |
| **VB–VS** | Bottom part of village Retje in a west-north direction. | 707.1–719.2 | 0.07 |
| **VS & SR** | Slope route in a village Retje-Hrib direction. | 720.3–758.4 | 0.16 |

**Figure 5: The location codes assigned to the line segments of the fixed route.**

eBC and PM$_{2.5}$ mass concentration variability trend differs throughout the route and it is day time dependent as well. Namely, mass concentrations of both pollutants are 2- to 4-times higher during evening temperature inversions as compared to inversions in the morning and in the afternoon. This applies to all locations inside the hollow. Yet, the difference between the run times is the highest in Retje village, with up to 5- and 7-times higher evening eBC and PM$_{2.5}$ levels, respectively.

Unlike the PM$_{2.5}$, the eBC trend along the line segment VB–BR depends on the time of the day. In the morning, from 6:30 to 9:00, eBC levels with the mean value of 3 µg m$^{-3}$ remain constant along the whole bottom route (BR), whereas around noon and in the early evening, eBC mass concentrations vary. As seen with the PM$_{2.5}$ levels, early evening eBC mass concentrations increase from 6.6 µg m$^{-3}$ to 11.6 µg m$^{-3}$ towards the village of Retje. In the other direction, from the village of Retje to st. Florjan chapel (VB–C), levels of both pollutants decrease, with the highest drop observed during evening temperature inversions, particularly in PM$_{2.5}$.

Horizontal gradients of the pollutants' mass concentrations differ the most at Retje village located at the bottom of the hollow
(VB–VS) and at the slope street in the direction from Retje village towards the settlement Hrib (VS & SR). This relates to morning and evening temperature inversion runs, while for the afternoon runs variability of the pollutants is similar. In the morning and in the evening, PM$_{2.5}$ mass concentrations over the whole village at the bottom of the Retje hollow are elevated and steeply decline further away from the village, especially in the evening. In the afternoon, however, greater mass concentrations were measured along the village on the slope (VS). This is seen in Fig. 4, line segment VB–VS, as a sharp
increase in PM$_{2.5}$ mass concentrations at a distance of 200 m away from the starting point of the route segment. The decrease of eBC along the slope road (VS & PC) from Retje village towards Hrib is smaller than that of PM$_{2.5}$. Moreover, during morning runs, eBC mass concentrations in the middle of the road segment increase, i. e. from 4.4 to 9.2 µg m$^{-3}$.

### 3.3 Mixing height during temperature inversions

The MH was determined for the selected temperature inversion events, based on eBC vertical profiles collected with mobile
measurements described under Sect. 2.3.2. Despite the method shortcomings, applied data quality measures ensure to meet the requirements of the study's purpose (see the following Sect. 3.3.1). The lowest point in the hollow captured by mobile measurements is at 705.9 m a.s.l. (the lowest elevation of the hollow is 700 m) while the highest point is at the top of Tabor hill at 815 m a.s.l (the highest point of the hollow's rim is at about 900 m a.sl.).

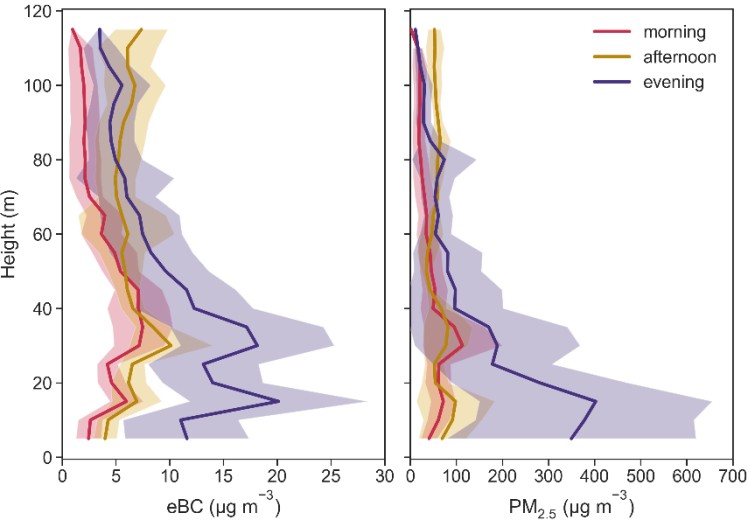

**Figure 6: Mean vertical eBC and PM$_{2.5}$ profiles with standard deviation (shaded area) during morning (6:30–8:30 LT, red line), afternoon (12.00:14.00 LT, yellow line) and evening (17:00–19:00 LT, purple line) winter-time temperature inversions over the Retje hollow.**
In Fig. 6 the 5 m averaged vertical profiles of eBC and PM$_{2.5}$ mass concentrations are depicted for all morning (6:30–9:00 LT), afternoon (12:00–14:00 LT) and evening (17:00–19:00 LT) runs during temperature inversions. Particle accumulation in the
490 mixing layer results in a strong concentration gradient indicating the MH. From eBC vertical profiles the mixing layer boundary is clearly visible as a sudden and significant decrease in eBC mass concentrations (all vertical profiles during temperature

inversions obtained with mobile runs are illustrated in the Supplement, Fig. S13). On average, during strong winter temperature inversion events (46 runs) the MH in the hollow is $58 \pm 15$ m. For temperature inversions with a stronger temperature gradient of about $4.7 \pm 1.7$ K (100 m)$^{-1}$, prevailing in December, the MH is on average 6 m higher ($61 \pm 15$ m) than during temperature inversions with a weaker temperature gradient of $2.5 \pm 2.1$ K (100 m)$^{-1}$ ($55 \pm 12$ m).

**Table 6. Calculated mean MH during all morning, afternoon and evening runs with ground temperature inversions, derived from eBC vertical mass concentration profiles (N=46). The MHs are rounded to 5 m.**

| Run period | Morning (6:30–8:30 LT) | Afternoon (12:00–14:00 LT) | Evening (17:00–19:00 LT) |
|---|---|---|---|
| eBC-MH average | $60 \pm 15$ m | $60 \pm 20$ m | $55 \pm 15$ m |

During stagnant atmospheric conditions a slight evolution in day-time MH is observed (Table 6. and Fig. 7.). The highest MH is detected during morning and afternoon runs with the mean height of about 60 m, whereas the lowest MH is in the early evening, with mean MH of 55 m. The highest variability of the mixing boundary layer height is observed around noon time with the MH in range between 40 to 105 m. As seen in Fig. 7, most of the points are positioned at 70 m MH, especially during morning runs. This corresponds to the height of drainage area at the south-eastern part of the hollow influenced by locally induced flows. Moreover, there is less emission sources above this height than in the lower parts of the hollow.

The MH increases during stronger temperature inversions in the morning and decreases during weaker temperature inversions at noon. Nonetheless, this information must be treated with caution since only seven afternoon runs were conducted, during which temperature inversion occurred.

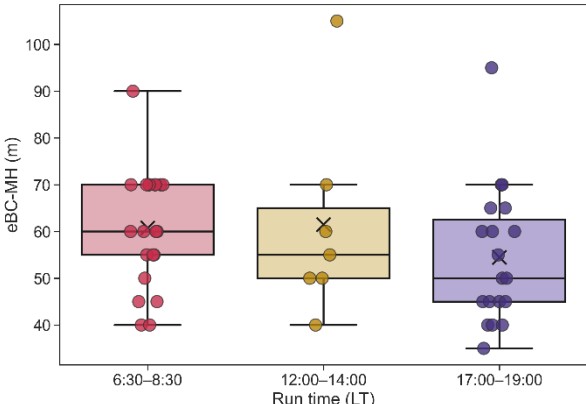

**Figure 7: Mixing height (MH) during morning (6:30–8:30 LT), afternoon (12:00–14:00 LT) and evening (17:00–19:00 LT) runs with temperature inversion. Statistics for every run time in a day are represented by a boxplot (cross: mean, horizontal line: median, box: 25$^{th}$–75$^{th}$ percentile, whiskers: 5$^{th}$–95$^{th}$ percentile).**

eBC mass concentrations within the MH (average of 8.2 µg m$^{-3}$) are almost 3 times higher than just above the MH in the free atmosphere (average of 3.2 µg m$^{-3}$). On average, free atmosphere represents 17 % of the total concentrations within the mixing height and 14 % of the total concentrations in the whole measured vertical profile. The lowest eBC mass concentrations during temperature inversions were measured during morning runs. The mean eBC mass concentrations detected in the whole vertical profile were $4.0 \pm 4.2$ µg m$^{-3}$, within the MH $5.0 \pm 4.5$ µg m$^{-3}$ and above MH $1.7 \pm 1.6$ µg m$^{-3}$. Early evening runs are

characterized by the highest eBC mass concentrations with mean eBC levels of 12.3 ± 8.9 μgm$^{-3}$ within the mixing height, 4.2 ± 3.4 μg m$^{-3}$ above MH and 9.8 ± 8.5 μg m$^{-3}$ in the whole vertical profile. During afternoon runs the difference between mass concentrations within the MH and just above the MH was 2 times smaller than during morning and evening runs. The mean eBC value within the MH was 6.9 ± 5.0 μg m$^{-3}$ compared to 5.1 ± 3.7 μg m$^{-3}$ of eBC detected just above the MH. On average, eBC mass concentrations in the free atmosphere represented 32 % of the total eBC mass concentrations within the mixing height.

Besides the strong eBC mass concentration gradient, representing the border of the MH, two other layers of high eBC mass concentrations are seen in most of eBC height profiles (Supplement Fig. S13). As seen on the spatial maps (Fig. 3.) the first increase of eBC as well as PM$_{2.5}$ levels arise close to the hollow's floor and coincides with the elevation of the village Retje at around 710 m a.s.l (VB). The second increase occurs on populated slope (SR), facing south, between 725 m and 745 m a.s.l. In the early evening the highest eBC mass concentrations are observed near the bottom of the hollow (VB) reaching 22 μg m$^{-3}$. At that time PM$_{2.5}$ mass concentrations at the Retje village reach maximum mass concentrations as well, reaching 560.7 μg m$^{-3}$.

### 3.3.1 Limitations of the mixing height determination

Validation of the MH method is important due to robustness of the pseudo-vertical mobile measurements. We performed measurements along the circular fixed route, and therefore, passing different microenvironments of the hollow, we have reduced the direct effects of local emission sources. With measurements along the other unpopulated north-facing slopes of the hollow, we would not obtain the relevant vertical profile. This side of the hollow is covered with forest, has different terrain configuration and, moreover, would not provide sufficient answers in relation to the focus of our study (local air quality).

As mentioned in Sect. 2.3.2, the eBC-derived MH was compared to the θ-MH during temperature inversions in December (Fig. S14). The difference between group means is statistically significant, by about 0.3 of standard deviations lower eBC-MH compared to θ-MH. Population value of a slope between the θ-MH and eBC-MH, is likely to fall between 0.52 and 0.92 (95 % confidence interval). The relatively wide interval suggests a high degree of uncertainty of our method. Yet, we cannot conclude that an effect is important because the p-value, from which we determine significance, is affected by a small sample size (22 runs with both mobile temperature and eBC measurements). From frequency distribution of the θ-MH and eBC-MH data shown in the subplots of Fig. S14 in the Supplement it is evident that there is a peak in the number of runs with θ-MH of 60 m, while for the eBC-MH, most of the runs have MH of 40 and 70 m. The former eBC-MH represents predominantly early evening runs and the latter morning runs.

The difference between the θ-MH and particle/eBC-MH over complex terrain was observed in some other studies as well (e.g. Ferrero et al., 2010; De Wekker et al., 2012; Lang et al., 2015). Furthermore, Seibert et al. (2000) states that there is a lower agreement between the two methods for very low MH. The correlation coefficient of this study agrees well with the profile with timing and resolution comparability issues reported in Ferrero et al. (2011). Besides, root mean squared error (RMSE) indicates that our model misses actual MH values by less than 12 m (RMSE = 11.4 m). Mean absolute percentage error

(MAPE) of the linear model's predictions are, on average 12.3 % off from actual value. Moreover, there are more negative errors than positive, indicating a systemic underestimation of the eBC-MH.

Comparing the results shown in the Supplement Fig. S14 it is important to consider that time resolution of potential temperature measurements was lower (2 min) than the resolution of particle measurements (10 s), which leads to the reduced accuracy of the θ-MH. Namely, in 2 min time, approx. 12.5 m distance and minimum 2 and maximum 20 m relative height had been walked from Hrib up to Tabor hill (13.5 m on average), indicating that an θ-MH error could be as high as 20 m. Nonetheless, this is still within recommended resolution range for MHs lower than 250 m (Seibert et al., 2000). The comparability between the two methods is lower during weak inversions and during MHs with stratified PBL structure (mainly early evening runs). This coincides with findings of studies described in Seibert et al. (2000), Gregorič et al. (2020) and Ferrero et al. (2012). An important contribution to the uncertainty of the method arises from the route specific measurements. Activity of emission sources along the fixed route, mainly residential wood burning (RWC) and also morning traffic, have an impact on the obtained vertical eBC profile and thus, on the determined MHs. In addition, the influence of land use and cover, the terrain configuration with related slope winds have a stronger impact as opposed to the standard" vertical measurement method. All of these factors are interconnected and thus, their separate contribution in the frame of this study is not possible. For safe interpretation additional measurements would be needed, i.e. traffic counts, wind measurements and/or pseudo-vertical measurements along the other slopes of the hollow.

With the following steps we tried to address some of the listed challenges arising from vertical mobile measurements along the slopes. Firstly, single events (e. g. construction work, chimney plume or passing a heavy-duty vehicle) were minimized by averaging 1-s raw data of AE51s to 10-s medians. The vertical eBC profiles presented in the study are an average of many measurements rounded to the nearest 5 m relative height. Hence, the effect of the AE51s measurement noise on the reported mass concentrations is further reduced. The effect of data aggregation is demonstrated by the standard deviation of the vertical eBC mass concentrations in Fig. S13. It is evident that measurements in areas with a higher concentration level have a higher variability than in the lower levels. Moreover, the special single events, which could have biased the results, were noted in the mobile measurements log-book and taken into consideration. Secondly, the obtained vertical profile of the hollow is a result of data points of different parts along the hollow. There is an overlap between different line segments and thus, up to 60 m relative heights, the profile is not route part specific (see Fig. S11 for the route parts height ranges). However, from there to the top of the hill at 115 m height, there is only one line segment (part of H and TR). With crossing the latter at least twice per every run (up and down the hill and after completion of some runs once more) we averaged out the single events and obtained representative distribution of eBC concentrations for the south east side of the studied area. Yet, we cannot claim that the results are valid for the other unpopulated south side of the hollow. Finally, for the MH determination, we have used data from the fixed stations as well (see Fig. S13). Besides θ, we considered RH measured at the three stations (Retje village station, meteorological station Hrib and Tabor background station). A significant reduction in RH among those stations was used as a sign for the height of a mixing layer. Additional control parameter, whether the MH is below or above the Tabor hill, was a comparison between eBC mass concentrations at the Retje village station and at the rural background station on Tabor hill. As

shown also in the study of Gregorič et al. (2020), higher concentrations in terrain depression than on the top of hill indicate MH below the altitude of the hill site.

### 3.4 Association between meteorological variables with eBC and PM at the fixed stations

Correlation analysis between hourly and daily meteorological variables and eBC mass concentrations at both air quality stations revealed a statistically significant correlation between eBC levels and temperature, HDD, ambient pressure, wind
speed and the potential temperature gradient. No statistically significant association was found in the following meteorological variables: wind direction (N=73) and daily sum of precipitation (N=38). The conditions for correlation analysis between MH and pollution levels or any other meteorological parameter have not been fulfilled, as the determination of MH was done only for the runs during temperature inversions (N=47) with a limited height range between 25 to 105 m, mostly from 40 to 75 m. A stronger correlation between most of the variables is found at the Retje village site, located in the hollow, than at the Tabor
rural background station, positioned on top of the hill. The only exceptions were the wind speed measurements performed at the meteorological station Hrib, positioned outside the hollow. At the village station correlation analysis with 24-hour $PM_{10}$ samples was conducted as well. Very similar correlation coefficients were obtained as for 24-hour average eBC mass concentrations and are presented in Table 7.





**Table 7: Regression results for the relation between meteorological conditions, eBC and PM$_{10}$ mass concentrations (µg m$^{-3}$) (24-hour averages) in winter 2017–2018 and for selected days with temperature inversion and unstable atmosphere (in brackets) at both air quality stations – the rural village station Retje (eBC and PM$_{10}$) and the rural background station Tabor (eBC). In the last column mean of meteorological parameters with standard deviation (std) are given.**


| Meteorological parameter | Pollutant at the site | slope | $R^2$ | p | N | mean ± std of meteorological parameter |
|---|---|---|---|---|---|---|
| **T (°C)** | eBC Retje | -0.51 ± 0.07 (-0.58 ± 0.09) | **0.39** **(0.42)** | <0.001 | 80 (57) | -0.6 ± 6.3 (-1.3 ± 6.4) |
| | PM$_{10}$ Retje | -5.79 ± 0.74 (-6.43 ± 0.92) | **0.43** **(0.47)** | <0.001 | 80 | |
| | eBC Tabor | -0.10 ± 0.03 (-0.12 ± 0.04) | **0.12** **(0.12)** | <0.001 | 90 (65) | -0.5 ± 5.5 (-1.0 ± 5.6) |
| **RH (%)** | eBC Retje | - (-0.27 ± 0.13) | **-** **(0.07)** | - (<0.05) | - (57) | 85.8 ± 9.9 (86.1 ± 9.2) |
| | PM$_{10}$ Retje | - (-3.25 ± 1.38) | **-** **(0.09)** | - (<0.05) | - | |
| | eBC Tabor | -0.05 ± 0.02 (-0.11 ± 0.03) | **0.05** **(0.16)** | <0.05 (<0.005) | 90 (57) | 85.8 ± 11.5 (86.2 ± 11.0) |
| **v* (m/s)** | eBC Retje | -2.28 ± 0.48 (-2.65 ± 0.62) | **0.24** **(0.28)** | <0.001 | 73 (50) | 0.9 ± 0.91 (0.96 ± 1.4) |
| | PM$_{10}$ Retje | -22.73 ± 4.89 (-26.80 ± 6.33) | **0.22** **(0.27)** | <0.001 | 79 (50) | |
| | eBC Tabor | -0.71 ± 0.14 (-0.78 ± 0.19) | **0.25** **(0.27)** | <0.001 (<0.001) | 79 (50) | |
| **wd* (°)** | eBC Tabor | - | **-** | - | - | SE–E |
| **p* (hPa)** | eBC Retje | 0.30 ± 0.04 (0.32 ± 0.05) | **0.45** **(0.47)** | <0.001 | 80 (57) | 1016.8 ± 9.5 (1017.9 ± 10.7) |
| | PM$_{10}$ Retje | 3.11 ± 0.40 (3.34 ± 0.49) | **0.42** **(0.46)** | <0.001 | 88 (57) | |
| | eBC Tabor | 0.10 ± 0.00 (0.11 ± 0.01) | **0.47** **(0.53)** | <0.001 | 90 (65) | |
| **H$_p$ (mm)** | eBC Retje | - | **-** | - | 32 (23) | 3.5 ± 7.7 (4.0 ± 8.8) |
| | PM$_{10}$ Retje | - (-2.24 ± 10.75) | **-** **(0.19)** | - (<0.005) | 38 (23) | |
| | eBC Tabor | - | **-** | - | 38 (23) | |
| **Θ grad (K (100 m)$^{-1}$)** | eBC Retje | 1.98 ± 0.13 (2.07 ± 0.14) | **0.75** **(0.8)** | <0.001 | 82 (59) | 1.6 ± 1.9 (1.8 ± 2.2) |
| | PM$_{10}$ Retje | 20.21 ± 1.42 (21.54 ± 1.50) | **0.75** **(0.79)** | <0.001 | 82 (57) | |
| | eBC Tabor | 0.55 ± 0.06 (0.56 ± 0.06) | **0.53** **(0.59)** | <0.001 | 82 (59) | |

*Data from the station Hrib (775 m a.s.l.).

At both sites the relation between the temperature and eBC mass concentrations is confirmed at the p < 0.001. On the other hand, only a very weak correlation ($R^2$=0.2, slope=-0.1) is found at the Tabor rural background station and a little bit higher, at the Retje village station, with $R^2$ amounting to around 0.4. The same influence is also observed for $PM_{10}$. The relationship

is negative, meaning that eBC and $PM_{10}$ mass concentrations decrease with higher temperatures, on average by -0.51 µg m$^{-3}$ and 5.79 µg m$^{-3}$, respectively. The results, however, are not most reliable as the correlation is moderate. The same relationship is found for HDD (Eurostat, 2019; ARSO, 2020). According to the HDD index there was the need for indoor heating in all winter days.

Relation analysis of wind speeds and eBC was performed, which considered only days with no precipitation and wind speeds

higher than 0.09 m s$^{-1}$. The location of the wind measurements is not representative of the conditions on the floor of the hollow. Therefore, correlation analysis is adequate only for the station on Tabor hill. However, merely a moderate correlation was found for the winter days. The relation with eBC is negative with a Pearson correlation coefficient R of -0.5. On average there is a reduction of 31 % in eBC per 1 m s$^{-1}$ at Tabor hill, yet the results cannot be used with confidence due to the relatively low $R^2$. Among all meteorological variables the strongest and the highest correlation was found between the potential temperature

gradient in the hollow and pollutants mass concentrations ($R^2 > 0.75$). A particularly high correlation ($R^2 = 0.8$) is determined at the Retje village station during the selected temperature inversion days and non-inversion days. eBC and $PM_{10}$ mass concentrations increase with a higher potential temperature gradient of the hollow, on average by 1.98 µg m$^{-3}$ and 20.21 µg m$^{-1}$ per 1 K (100 m)$^{-1}$, respectively. This increment represents the largest induced change in pollution levels of all the meteorological parameters studied.

**3.4.1 Correlation analysis inside the cold air pool**

High time and spatial resolution data of mobile measurements allowed us to explore the spatiotemporal dependency of the eBC and $PM_{2.5}$ mass concentrations along the hollow to the meteorological variables. The linear correlation between the variables studied for the different run times is shown in Fig. 8.

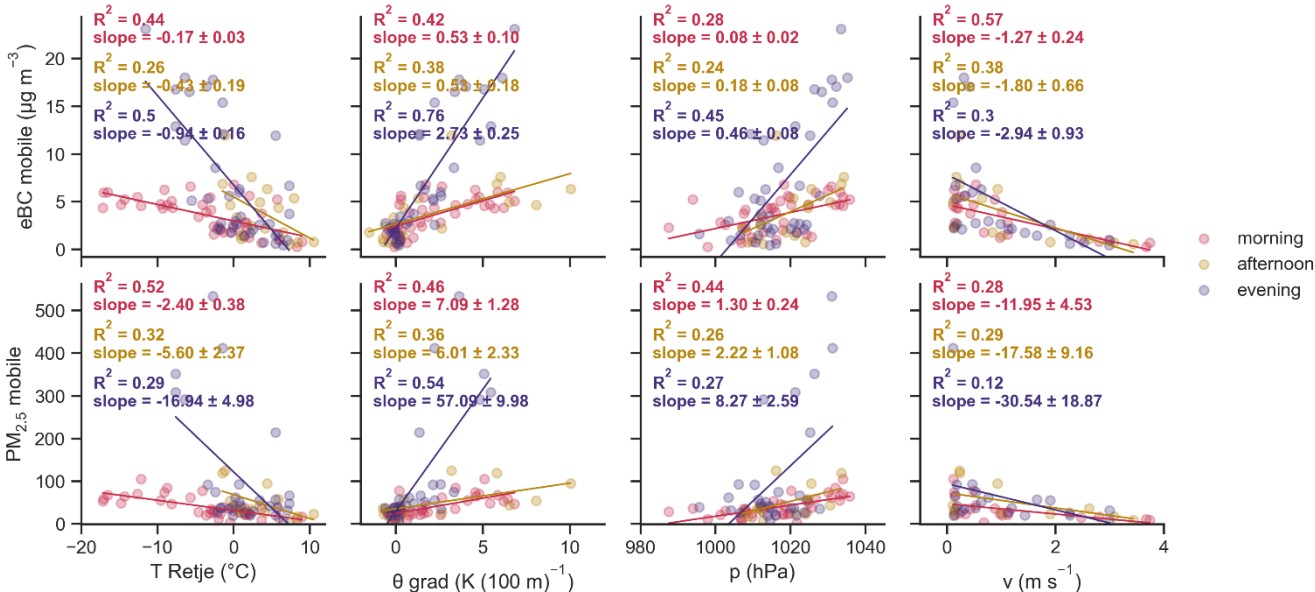

Figure 8: Linear correlations between the temperature measured at the Retje station (T Retje), potential temperature gradient ($\theta$ grad), atmospheric pressure (p) and wind speed measured at the Hrib (v) with mobile eBC and PM$_{2.5}$ mass concentrations inside the cold air pool (CAP).

Temperature dependence of eBC is higher for evening runs ($R^2 = 0.5$) compared to morning ($R^2 = 0.44$) and afternoon ($R^2 = 0.26$) runs. However, for PM$_{2.5}$, the correlation is stronger in the morning ($R^2 = 0.5$) than in the evening and in the afternoon ($R^2 = 0.3$). Yet, the gradient of the linear regression line is, like eBC, the steepest in the evening. With temperature decrease of -1 °C, eBC and PM$_{2.5}$ are reduced by 14 %.

Despite not the most representative location of wind measurements for the atmospheric conditions inside the CAP, wind-speed dependence of eBC and PM$_{2.5}$ obtained with mobile measurements was confirmed as well. Correlation is better for eBC than for PM$_{2.5}$ and it is the strongest for the morning runs ($R^2 = 0.6$ for eBC and $R^2 = 0.3$ for PM$_{2.5}$). Nonetheless, the reduction in pollutants levels in the hollow per 1 m s$^{-1}$ of wind speed is higher in the evening (-40 %) compared to afternoon and morning runs (from -22 to -30 %).

Unlike the other meteorological variables, the correlation between the pollutant levels and the potential temperature gradient of the hollow is the strongest in the early evening with $R^2 = 0.8$ for eBC and $R^2 = 0.5$ for PM$_{2.5}$. During this time, the increase in the pollutants levels is also by 63 to 72 % higher compared to the morning and the afternoon runs. The change in pollutant mass concentrations in the evening is comparable to those induced by wind speed. However, the significance of the wind speed model is low with $R^2$ from 0.12 to 0.3.

## 4 Discussion

We found out that out of 37 M people in Europe who live in rural mountain areas, 80 % of them live in relief depressions, most of them being from Central Europe. Yet only little is known about the real air pollution levels in those areas. Similar to many other locations in the Alps (Herich et al., 2014) and many other rural hilly and mountainous areas in Europe (e.g. Reis et al., 2009; Wählin et al., 2010; Becerril-Valle et al., 2017; Chevrier, 2017), residential wood burning is a major source of airborne particles in the study area. This was assessed with the so-called "Aethalometer approach" (Sandardewi et al., 2008) in previous research (Glojek et al., 2018; 2020), which is further confirmed by the low ratio of eBC to $PM_{2.5}$ observed in this study. Namely, the calculated ratios with values between 0.062 and 0.094, correspond to other ratios calculated in the areas with residential wood combustion and for forest fires (Herich et al., 2014; Liu et al., 2018). Moreover, the high correlation between eBC and $PM_{2.5}$ mass concentrations along the whole hollow indicates homogeneous atmospheric conditions with the prevalence of one emission source. However, different morning and evening horizontal gradients of eBC and $PM_{2.5}$ were observed during temperature inversions in the village of Retje. We partially interpret this as secondary aerosol formation (SOA).

Our measurements confirmed favourable topographical and meteorological conditions for the formation of temperature inversions in the studied hollow, particularly in the cold part of the year. These stable atmospheric conditions have a significant impact on the air pollution levels in the studied area. Therefore, eBC and $PM_{2.5}$ mass concentrations build-up in the whole hollow during temperature inversions and reached values (mean of $4.5 \pm 2.6 \ \mu g \ m^{-3}$ of eBC and $48.0 \pm 27.7 \ \mu g \ m^{-3}$ of $PM_{2.5}$) that are comparable to larger European centres (e.g. Fuller et al. 2014; Manigrasso et al., 2017; Küpper et al., 2018). Moreover, they were above the EEA yearly limit and WHO daily guideline values for $PM_{2.5}$ (both account for $25 \ \mu g \ m^{-3}$). According to the European Air Quality Index (EEA, 2020) for $PM_{2.5}$ and $PM_{10}$ mass concentrations levels, air quality in the hollow during temperature inversions was very poor. For the entire winter campaign 2017–2018, however, it was moderate.

During temperature inversion, pollution levels in the hollow were the greatest in early evenings, reaching up to $22 \ \mu g \ m^{-3}$ of eBC and $560.7 \ \mu g \ m^{-3}$ of $PM_{2.5}$. This is the result of domestic wood burning, which increases when people return home after work, and a very shallow mixing height of only 55 m on average, limited to the bottom of the hollow alone. However, during unstable atmosphere both eBC and $PM_{2.5}$ levels in the hollow dropped to less than $1 \ \mu g \ m^{-3}$ of eBC and $12 \ \mu g \ m^{-3}$ of $PM_{2.5}$, which is approximately 4 times lower than during temperature inversions and corresponds to European regional background sites values (EEA, 2019; Lilijana Kozlovič et al., 2019; Sun et al., 2019). However, in proximity of combustion sources, namely in the village Retje at the bottom of the hollow (VB) and at the settlement Hrib on the south-eastern slope, pollutant mass concentrations stayed above $1.8 \ \mu g \ m^{-3}$ of eBC and $12 \ \mu g \ m^{-3}$ of $PM_{2.5}$.

Despite the size of the hollow, covering an area of $1.5 \ km^2$ only, we demonstrated that an increase in pollutant mass concentrations during temperature inversions is not the same in the whole hollow. Moreover, it strongly differs within a 250-m radius (see Fig. 3. and 4.). Besides, maximum mass concentrations are not always associated with the location of local emission sources. This coincides with Gohm et al. (2009) findings, which reject the concept of spatial concentration

homogeneity under the mixing height in a complex terrain. Moreover, our results demonstrate that spatial inhomogeneity in small-size hollows could be even smaller than 1 km as observed in the study mentioned above. Spatial distribution of eBC and PM$_{2.5}$ mass concentrations differ during the day as well, largely because of different atmospheric conditions. During the morning runs with temperature inversion i.e. from 6:30 to 9:00 LT, the maximum of eBC as well as PM$_{2.5}$ occurred along the road on the lower slope (SR), facing the southern side of the hollow, and not at the bottom of the hollow (BV), where the highest number of emission sources were detected. This might have happened because the morning run along the slope route was conducted between 8:15 and 8:30 LT, when the sun had already risen. Radiative heating of air above the slope exposed to the sun with no or almost no snow, having a lower albedo compared to a snow-covered floor in the hollow, may be strong enough to induce upslope winds. Differences in temperature above different types of land covers in the hollow were previously detected by a thermal heat camera in the study of Vysoudil et al. (2019). This is one of the possible reasons for the highest pollution levels on the first slope road. However, to verify this hypothesis, wind measurements in the hollow would be required. Local morning traffic (people driving to a local school and to their workplace) contribute slightly as well, especially when it comes to eBC mass concentration levels (Glojek et al., 2020). Yet, the lowest eBC and PM$_{2.5}$ mass concentrations were detected at the end of the path, at the north-western most point of the hollow (C). The temperature and pollutant concentrations, however, remained the same at the bottom of the hollow. Contrary to pollutant maxima, the location of the minima might be explained with a drainage flow out of the hollow. On the way from the village of Retje towards the chapel of st. Florjan, the hollow is narrower with a slight increase in altitude close to the chapel. The speed of the cold air pool flow in the narrower part of the hollow towards the chapel is higher and the pressure is lower. The air flow creates an underpressure which sucks the air from the cold air pool at the bottom of the hollow to the southeast end of the hollow, i. e. the chapel of st. Florjan. As an outflow compensation, cleaner air from the nearby wooded surrounding inflows to the place. The location of the minimum pollutant level in the evening is the same. However, in the evenings with a lower mixing height (55 m on average), the maximum of eBC and PM$_{2.5}$ mass concentrations is limited to the bottom of the hollow alone. During the day thermal forcing, occurring due to solar irradiation, has the greatest impact on dispersion of pollutants. Like the morning runs, maximum pollution levels were reached at the start of the lower slope road (SR), facing south. Notwithstanding, in the afternoon the difference between eBC mass concentrations above the mixing height and the levels within the hollow was not so distinct. Moreover, eBC mass concentrations were increasing with the height, forming an elevated pollution layer. The formation of the elevated pollution layer by cross-valley flow and upslope wind transport was detected by previous researchers studying air pollution and meteorological processes in cold air pools as well (e.g. Gohm et al., 2009; Shapiro, Fedorovich, 2007; Rakovec et al., 2002). Connected to this, it should be noted that the settlement Hrib, positioned mostly above the mixing layer height of the studied hollow, has most likely a greater effect on pollutant levels on top of the hill Tabor than more distant sources from the village Retje at the bottom of the hollow.

The impact of temperature inversions on pollutant levels was additionally confirmed by a strong correlation between the eBC mass concentrations in the hollow and the vertical potential temperature gradient ($R^2 > 0.7$). The correlation was the highest for the selected temperature inversion and non-inversion days ($R^2 > 0.8$). Correlation coefficients ($R^2$) of around 0.7 to 0.9 were

also observed in other studies considering $PM_{2.5}$ or $PM_{10}$ pollution levels in winter and measures of atmospheric stability in relief depressions (e.g. Whiteman et al, 2014; Largeron, Staquet, 2016). A moderate correlation ($R^2$ of 0.5) between eBC and ambient pressure was detected as well since it is indirectly connected with the formation of ground temperature inversions. Because the measurement campaign was performed in the cold part of the year, correlation analysis of mean daily ambient temperature did not show a considerable impact on the pollution levels ($R^2=0.4$). Namely, all days in winter fulfilled the criteria to be classified as heating degree days (HDD according to Eurostat, 2019 and ARSO, 2020). However, the impact of temperature differs within a day. In the morning, when temperatures in the hollow drop to their lowest, the highest impact on $PM_{2.5}$ mass concentrations was observed. The effect on the eBC mass concentration is slightly lower, probably due to the increased morning traffic. As already described in the result section, wind measurements were performed outside the studied hollow and therefore did not show a very strong relation with eBC mass concentration levels in the hollow. Thus, wind speeds in the relief depression are expected to be even lower due to the topographical effect. Nonetheless, the decrease of pollutants mass concentrations with increasing wind speed in the hollow is clearly observed, indicating entirely local origin of pollution. It should be noted that the meteorological variables considered are often interrelated (e.g. precipitation and wind, temperature and RH) and can often not be treated individually.

## 5 Conclusions

Pilot study to determine the residential wood combustion air pollution under "real-world laboratory conditions" is presented. The studied example is representative for rural hilly and mountainous areas in central and southeastern Europe. We provide rare information on eBC and PM air pollution in rural small-scale relief depressions with mainly one emission source, i.e. residential wood burning. Due to its symmetrically shaped topography, smaller size (1.5 $km^2$ and less than 150 m deep) and remoteness, the selected Retje karst hollow in Slovenia is very convenient for the study of the impact of temperature inversions on local air quality.

Sampling with a mobile monitoring platform was conducted along the whole karst hollow in December 2017 and in January 2018. The mobile monitoring platform was equipped with portable instruments measuring at time resolution of 10 s. The monitoring sites with reference instruments (Aethalometer AE33 for eBC, Digitel DHA-08 and the TROPOS-type MPSS model for PM determination) were selected according to their position in the hollow and land use. The rural village station was located at the bottom of the karst depression whereas the rural background station was positioned at the top of the hill Tabor.

The primary emission source of eBC and PM in the area is residential wood combustion that is also indicated by a low ratio of eBC to $PM_{2.5}$, ranging from 0.062 to 0.094. Performed measurements confirmed meteorological conditions favourable for the formation of ground temperature inversions in the studied hollow. Namely, temperature inversion was present in more than 70 % of all winter nights and mornings. These very stable conditions prevent effective mixing and dispersion of pollutant concentrations in the hollow, leading to elevated pollution levels. Therefore, during temperature inversions eBC and $PM_{2.5}$

mass concentrations in the hollow increase to levels which are comparable to larger European urban centres and are above the EEA daily limit value for $PM_{10}$ i.e. 50 µg m$^{-3}$ as well as above yearly limit and WHO daily guideline values for $PM_{2.5}$, i.e. 25 µg m$^{-3}$.

Compared to larger and deeper basins and valleys, in small-size, shallow hollows the dynamic of temperature inversion seems to be much faster. Due to a smaller volume of the air in the hollow, a considerably lower number of emission sources can generate eBC and $PM_{2.5}$ pollution which is comparable to larger urbanised basins and valleys. Our measurements, therefore, highlight that eBC and PM levels in rural shallow terrain depressions with residential wood burning could be much greater than predicted by models. Besides, even in a small hollow as presented in this case study, pollutant concentrations differ
significantly depending on its location. During morning and afternoon temperature inversions people living at the lower part of south-facing slopes proved to be the most exposed to the high eBC and $PM_{2.5}$ mass concentrations, while in the early evenings, when the MH is limited only to the bottom of the hollow, people in the village Retje, living on the floor of the hollow breathe the highest pollutants level. Pollutant concentrations measured during temperature inversions in the rather sparsely populated small relief depression are a cause for concern since similar conditions can be expected in numerous hilly and
mountainous regions across Europe where approximately 20 % of the total population lives, out of which 30 % live in rural relief depressions, which is akin to the Retje site.

The results of this study highlight the importance of quality high-resolution air quality measurements for monitoring, assessing the effects of actions on the local air pollution, and reducing pollution by residential wood burning, especially in mountainous areas with limited self-cleaning capacities of the atmosphere. Therefore, we suggest the following:

1. to look at pilot sites on smaller spatial scale, which could help decision makers in taking effective measures at a local scale;

2. to raise awareness and knowledge about the air pollution problem of wood burning among the people, including capacity building about the negative health effects, energy efficiency, economical costs of ineffective combustion, optimal use, and regular maintenance of heating appliances, use of quality fuel (e.g. dry wood);

3. to inform the inhabitants, when atmospheric conditions impede effective dispersion and wood burning is not recommended;

4. to identify local super emitters, since they could be the main cause of deteriorated local air quality;

5. to upgrade existing stoves, enhance energy-related buildings renovations, and changing the fuel if a better alternative exists are seen as possible options to reduce wood burning pollution;

6. to actively involve the local community, taking measures of pollutant emissions reduction;

7. to highlight that there is no one single solution and to address the problem successfully, because multilevel actions are
790 needed, considering geographical and cultural characteristics of locations.

*Code and data availability.* The data used in this publication are available upon request to the corresponding author (k.glojek@gmail.com).

*Competing interests.* The authors declare that they have no conflict of interest.

*Author contributions.* AW, GM, LD, MO and KG conceived and designed the study. AW, HDCA and TM designed the mobile measurements and the aerosol backpack. KG, HDCA, KW, MMe, LD, GM, MO, DP, IJ and MRi installed, operated, maintained, calibrated and performed checks of the instruments before, during and/or after the campaign. KG, MR, MMa and AG performed field measurements with assistance from HDCA and KW. DP an HH provided data of the filter-based measurements. KG processed the data and prepared the manuscript with inputs from AG, HDCDA, ACM, LD, MO, TM, MR, DP, MRi, MMa and AW. All authors contributed to the scientific discussion and approved the manuscript.

*Acknowledgements.* The authors acknowledge the financial support from the Slovenian Research Agency (program MR-2016, program P1-0385 "Remote sensing of atmospheric properties"), Municipality of Loški Potok, and the COST Action CA16109 COLOSSAL. We would like to thank all the people and institutes involved in the campaign. We are truly grateful to all volunteers for the help with the mobile measurements and to the local community for their friendly welcome, help and support.

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
