# Peer review of "The impact of temperature inversions on black carbon and particle mass concentrations in a mountainous area"

_Atmospheric Chemistry and Physics, 2021_

## Referee Comment (RC1)

Comments to the manuscript: "**The impact of temperature inversions on black carbon and particle mass concentrations in a mountainous area**" by Glojek et al.

This manuscript aims at studying the influence of ground temperature inversions on air pollutants concentration in a hilly rural area (Retje karst hollow, Slovenia) impacted by wood combustion aerosols. eBC, PN and PM mass concentration measurements were performed at two air quality stations in the area of study (one placed at the bottom of the depression [Retje village] and one at the top of the hill [Tabor site]). Moreover, measurements were also performed by means of a mobile monitoring platform along a 6 km long route along the hollow.

Mobile eBC concentrations and temperature measurements along the hollow (from Retje (715 m a.s.l.) to Tabor (815 m a.s.l.)) and potential temperature measurements at three stations at different elevation (Retje: 715 m a.s.l., Hrib: 775 m a.s.l., Tabor: 815 m a.s.l.) were used to determine the inversion days and inversion heights by means of the gradient method.

The experiment providing the data for this manuscript was well designed and the experimental approach robust and well described. My detailed comments below.

**Comments:**

- **Comments on Introduction:**
- I agree with the authors about the difficulties that air quality models have in predicting dispersion and pollutant concentrations in complex relief characterized by strong inversions and strong stability of the lower atmosphere. I have seen strong underestimations of pollutant concentrations by models in such environments. Another feature of these atmospheric conditions is that these can increase the toxicity of ambient air through a progressive accumulation of PM (mostly primary), gaseous compounds, etc. The authors may want to cite a paper where the authors demonstrated that the MH is a key parameter to take into account since it is able to influence all-cause daily mortality more than PM (*Pandolfi et al., Science of the Total Environment 494–495, 283–289, 2014, http://dx.doi.org/10.1016/j.scitotenv.2014.07.004*).

- **Comments on Figure S13:**
- For the mobile measurements (December runs) the profiles of the potential temperature (and not of the actual temperature as it seems from the Figure S13/caption) should be reported in the Figure (light blue vertical lines). Moreover, it is not clear if T-MH (blue horizontal lines) represents the largest positive gradient of the potential temperature or of the actual temperature.
- The largest positive gradient of the potential temperature profile can be used for MH determination, as the potential temperature is higher above the inversion than below (positive gradient). However, it seems that T-MH (blue horizontal lines) is sometimes associated with negative gradients. For example: 171218C or 171224A profiles. Moreover, sometimes T-MH is associated (apparently) to a region where the temperature (light blue vertical line) is constant with height (for example: 171219B and 171225B profiles). Also, it seems that the T-MH was not calculated for some of the vertical profiles (for example: 171223C and 171231A profiles).

- If the gradient method was used for MH determination, then the vertical profiles of the T and eBC first derivatives should be also presented in supporting material.
- Three eBC-MH horizontal lines should be reported in the Figure (black, dark grey and light grey horizontal lines). However, I only see two of them.
- I do not understand how the final MH (red line) was obtained.

- **There are two Figures S12 in supporting material**

- **Comments to Table 5, Figure 5 and related text:**
- In Table 5 the reported MHs were retrieved using the eBC vertical profiles (negative gradient) from selected morning, afternoon and evening runs. Are these MH heights the same that were represented as red horizontal lines in Figure S13? Moreover, it is surprising to me that the authors were able to appreciated 5 meters difference between the evening runs and the other runs. Figures S12 (Correlation between T-MH and eBC-MH for temperature inversion runs in December) clearly shows that the correlation between eBC-MH and T-MH is not very high and that a bias is present. My main question is: how can the authors use eBC vertical measurements to determine the MH if eBC emissions can occur locally along the slope? The bias between eBC-MH and T-MH is actually quite high. Moreover, in Figure 3 it can be seen that eBC concentrations during inversions are higher along the hill than in Retje (pag. 16, lines 404-406).

- **Comments to section 2.3.1:**
- In this paragraph (and in many other part of the manuscript) the term "temperature" is used instead of "potential temperature". Please, change the text accordingly.
- The authors may want to cite a paper were the first derivative of the potential temperature vertical profiles was used to study how the unstable, stable or neutral atmospheric conditions of the atmosphere alter the distribution of aerosol backscatter ($\propto$ PM concentration) with height (*Pandolfi, M., Martucci, G., Querol, X., Alastuey, A., Wilsenack, F., Frey, S., O'Dowd, C. D., and Dall'Osto, M.: Continuous atmospheric boundary layer observations in the coastal urban area of Barcelona during SAPUSS, Atmos. Chem. Phys., 13, 4983–4996, https://doi.org/10.5194/acp-13-4983-2013, 2013*).
- I do not agree with the nomenclature used. I agree with the authors if they classify a run as "stable atmosphere" when the potential temperature increased with altitude along the depression. However, the authors defined the atmosphere as "mixed" when the potential temperature decreased (negative gradient) with height. In the presence of a negative gradient the atmosphere should be defined as "unstable" and not as "mixed". By definition a mixed atmosphere defines a region (the mixing layer) where the potential temperature is constant with height ($d\theta/dz = 0$). However, looking at Figure 2, the periods between two inversion periods (shaded areas) are characterized by equal (or very similar) potential temperature values at Tabor and Retje, thus indicating actually a mixed atmosphere (i.e. with $d\theta/dz = 0$).

- **Comments to section 2.3.2:**
- The authors defined the height that separates the boundary layer from the free troposphere (FT) as MH (mixing height) irrespective of the time of the day. I would rather use, for example, PBLH (planetary boundary layer height) as a more general definition of the height separating PBL and FT as MH is usually referred as the PBLH when the atmosphere is well mixed (midday).
- Given that the estimation of the eBC-MH could be hampered by eBC amissions along the slope, I would suggest the authors to compare eBC-MH also with AH-MH (AH = absolute humidity) during mobile run measurements.
- My suggestion is to include in this work the AH too (of course if RH measurements are available during mobile measurements).

- **Comments to Figure 2:**
- As in my previous comment. Looking at Figure 2, the periods between two temperature inversion periods (shaded areas) were characterized by equal (or very similar) potential temperature values at Tabor and Retje, thus indicating actually a mixed atmosphere (i.e. with $d\theta/dz = 0$). However, in the main text the authors defined the atmosphere during these periods as periods with $d\theta/dz < 0$ (whereas it seems from Figure 2 that $d\theta/dz = 0$). So, I have not understood how the authors defined a "mixed" atmosphere: when $d\theta/dz = 0$ or when $d\theta/dz < 0$?
- Interestingly, no rain was observed during temperature inversion periods. Very often, rain was detected just before and just after the inversion periods (shaded areas in Figure 2). Any comment about this?
- During stable atmospheric conditions the eBC concentrations were higher than usual at the top of the hill too (Tabor site). Was this relative increase due to some eBC transport from below (e.g. Retje)? or was due to eBC sources along the hill accumulated during stable conditions?

---

## Author Response (AR1)

**REVIEWER 1:**
Comments to the manuscript: "**The impact of temperature inversions on black carbon and particle mass concentrations in a mountainous area**" by Glojek et al.

*This manuscript aims at studying the influence of ground temperature inversions on air pollutants concentration in a hilly rural area (Retje karst hollow, Slovenia) impacted by wood combustion aerosols. eBC, PN and PM mass concentration measurements were performed at two air quality stations in the area of study (one placed at the bottom of the depression [Retje village] and one at the top of the hill [Tabor site]). Moreover, measurements were also performed by means of a mobile monitoring platform along a 6 km long route along the hollow. Mobile eBC concentrations and temperature measurements along the hollow (from Retje (715 m a.s.l.) to Tabor (815 m a.s.l.)) and potential temperature measurements at three stations at different elevation (Retje: 715 m a.s.l., Hrib: 775 m a.s.l., Tabor: 815 m a.s.l.) were used to determine the inversion days and inversion heights by means of the gradient method.*
*The experiment providing the data for this manuscript was well designed and the experimental approach robust and well described. My detailed comments below.*

We truly appreciate the comments and suggestions on the manuscript you have provided. Below you can find the responses to your comments. The text in blue presents the new text in the manuscript.

**Comments:**
- **Comments on Introduction:**
*- I agree with the authors about the difficulties that air quality models have in predicting dispersion and pollutant concentrations in complex relief characterized by strong inversions and strong stability of the lower atmosphere. I have seen strong underestimations of pollutant concentrations by models in such environments. Another feature of these atmospheric conditions is that these can increase the toxicity of ambient air through a progressive accumulation of PM (mostly primary), gaseous compounds, etc. The authors may want to cite a paper where the authors demonstrated that the MH is a key parameter to take into account since it is able to influence all-cause daily mortality more than PM (Pandolfi et al., Science of the Total Environment 494–495, 283–289, 2014, http://dx.doi.org/10.1016/j.scitotenv.2014.07.004).*

**Response:**

Thank you for suggestion. We have included the provided reference in the introduction.

**Changes in the manuscript:**

Line 64-77: »Consequently, air pollution problems in populated relief depressions, especially in small-scale ones, are often underestimated. Besides, in the study by Pandolfi et al. (2014), higher risk of mortality was observed during reduced mixing layer height, caused by the accumulation of anthropogenic primary air pollutants. In order to accurately investigate aerosol pollution processes in smaller areas with complex relief, high resolution quality measurements are needed. Available methods .... «

- **Comments on Figure S13:**
*- For the mobile measurements (December runs) the profiles of the potential temperature (and not of the actual temperature as it seems from the Figure S13/caption) should be reported in the Figure (light blue vertical lines). Moreover, it is not clear if T-MH (blue horizontal lines) represents the largest positive gradient of the potential temperature or of the actual temperature.*

*- The largest positive gradient of the potential temperature profile can be used for MH determination, as the potential temperature is higher above the inversion than below (positive gradient). However, it seems that T-MH (blue horizontal lines) is sometimes associated with negative gradients. For example: 171218C or 171224A profiles. Moreover, sometimes T-MH is associated (apparently) to a region where the temperature (light blue vertical line) is constant with height (for example: 171219B and 171225B profiles). Also, it seems that the T-MH was not calculated for some of the vertical profiles (for example: 171223C and 171231A profiles).*

**Response:**

The graphic presentation was changed as the light blue line was presenting temperature and not potential temperature. θ-MH was calculated for all vertical profiles with mobile temperature measurements but in cases (e. g. 171223C and 171231A ) where θ-MH matched the eBC-MH, only eBC-MH line was shown. In the updated Figure (see below), all lines are shown from which the red, boldest one, presents the final determined MH.

**Changes in the manuscript:**

*Upgraded Fig. S13 in the Supplement. Change of MH line indications and conversion of mobile measurements of temperature to potential temperature. eBC (black dots) and RH (dots in wine red) measured at the stations were added as well and standard deviation of the mobile eBC vertical profile (shaded grey area).*

[Figure]

**Figure S13: Selection of the runs with temperature inversion according to potential temperatures (θ) at three stations at different elevation (Retje: 715 m a.s.l., Hrib: 775 m a.s.l., Tabor: 815 m a.s.l.). Vertical 5 m averaged eBC (black line) and θ profiles (light blue line) from bottom of the Retje karst hollow (705 m a.s.l.) to the top of Tabor hill (815 m a.s.l.). eBC-MH1 (dashed black horizontal line) represent the largest negative eBC gradient, eBC-MH2 (dotted black horizontal line) second largest negative gradient, eBC-MH3 (dotted dark grey horizontal line) third largest negative gradient and final MH (red bold line) depicts selected MH according to eBC profile. For December runs MH was determined according to vertical θ profile as well (θ-MH, blue horizontal line). The largest positive gradient was selected as θ-MH.**

*- If the gradient method was used for MH determination, then the vertical profiles of the T and eBC first derivatives should be also presented in supporting material.*

**Response:**

We understand the added value of this information. We have added a table and additional text in the Supplement.

**Added to the manuscript:**

*Added Table in the Supplement.*

**Table S1: The θ (K (100 m)$^{-1}$) and eBC (μg m$^{-3}$) gradients at the determined MHs (in brackets)**

| Run id | 171204 A | 171204 C | 171205 A | 171205 B | 171205 C | 171206 A | 171206 C | 171209 C | 171210 A | 171216 C |
|---|---|---|---|---|---|---|---|---|---|---|
| Θ grad | – | – | – | – | – | – | – | – | – | +1.18 (20 m) |
| eBC grad | -6.48 (70 m) | -4.66 (70 m) | -10.57 (40 m) | -7.35 (70 m) | -8.88 (65 m) | -1.31 (50 m) | -5.79 (95 m) | -3.06 (70 m) | -1.49 (70 m) | -2.05 (35 m) |
| Run id | 171217 A | 171217 C | 171218 A | 171218 B | 171218 C | 171219 A | 171219 B | 171219 C | 171221 A | 171222 A |
| Θ grad | +3.20 (60 m) | +3.51 (55 m) | +1.72 (70 m) | +4.19 (105 m) | +2.92 (55 m) | +2.46 (50 m) | +0.89 (55 m) | – | +4.74 (85 m) | +1.40 (100 m) |
| eBC grad | -1.92 (45 m) | -5.53 (55 m) | -3.02 (70 m) | -3.12 (105 m) | -14.26 (45 m) | -1.94 (60 m) | -3.01 (60 m) | -18.78 (40 m) | -1.21 (70 m) | -0.87 (90 m) |
| Run id | 171222 C | 171223 A | 171223 B | 171223 C | 171224 A | 171225 B | 171225 C | 171229 A | 171229 C | 171230 A |
| Θ grad | +3.01 (65 m) | +1.98 (60 m) | +3.64 (60 m) | +2.43 (60 m) | +3.64 (70 m) | +1.14 (55 m) | +0.55 (55 m) | +2.66 (95 m) | +2.14 (40 m) | – |
| eBC grad | -2.07 (60 m) | -2.87 (55 m) | -2.13 (50 m) | -4.70 (60 m) | -5.35 (60 m) | -1.30 (50 m) | -1.00 (45 m) | -1.46 (70 m) | -4.58 (45 m) | -1.77 (55 m) |
| Run id | 171230 B | 171230 C | 171231 A | 180104 A | 180104 B | 180104 C | 180105 A | 180107 A | 180107 C | 180108 C |
| Θ grad | +3.66 (70 m) | +1.10 (60 m) | +1.19 (65 m) | – | – | – | – | – | – | – |
| eBC grad | -4.00 (55 m) | -4.98 (45 m) | -4.63 (45 m) | -1.11 (60 m) | -1.18 (40 m) | -2.52 (50 m) | -2.17 (55 m) | -2.70 (60 m) | -2.34 (50 m) | -1.38 (65 m) |
| Run id | 180118 A | 180120 A | 180120 C | 180122 C | 180123 A | 180123 C | 180124 A | | | |
| eBC grad | -3.31 (70 m) | -3.09 (70 m) | -2.23 (60 m) | -2.42 (40 m) | -3.36 (40 m) | -1.45 (40 m) | -7.45 (70 m) | | | |

*- Three eBC-MH horizontal lines should be reported in the Figure (black, dark grey and light grey horizontal lines). However, I only see two of them.*

**Response:**

See our response two indents higher.

*- I do not understand how the final MH (red line) was obtained.*

**Response:**

Thank you for this comment. We agree that from the Figure S13 this is not very clear. We have changed the Figure and added some explanation (text in blue) in the Supplement.

As we wrote in the manuscript, the final MH (red line) represents the largest negative eBC gradient, except in some cases when anomalies in the eBC profile were detected. For example, bias due to single events (e. g. increased traffic), change of the weather pattern during the run or small gradients along the profile. These exceptions are provided and described below the Figure S13.

**Changes in the Supplement text:**

**Figure S13: Selection of the runs with temperature inversion according to potential temperatures (θ) at three stations at different elevation (Retje: 715 m a.s.l., Hrib: 775 m a.s.l., Tabor: 815 m a.s.l.). Vertical 5 m averaged eBC (black line) and θ profiles (light blue line) from bottom of the Retje karst hollow (705 m a.s.l.) to the top of Tabor hill (815 m a.s.l.). eBC-MH1 (dashed black horizontal line) represent the largest negative eBC gradient, eBC-MH2 (dotted black horizontal line) second largest negative gradient, eBC-MH3 (dotted dark grey horizontal line) third largest negative gradient and final MH (red bold line) depicts selected MH according to eBC profile. For December runs MH was determined according to vertical θ profile as well (θ-MH, blue horizontal line). The largest positive gradient was selected as θ-MH.**

- **There are two Figures S12 in supporting material**

**Response:**

Thank you for this comment. We do see oversight and we have changed wrong Figure S12 into S11 in line with the pdf* document.

- **Comments to Table 5, Figure 5 and related text:**

*- In Table 5 the reported MHs were retrieved using the eBC vertical profiles (negative gradient) from selected morning, afternoon and evening runs. Are these MH heights the same that were represented as red horizontal lines in Figure S13? Moreover, it is surprising to me that the authors were able to appreciated 5 meters difference between the evening runs and the other runs. Figures S12 (Correlation between T-MH and eBC-MH for temperature inversion runs in December) clearly shows that the correlation between eBC-MH and T-MH is not very high and that a bias is present. My main question is: how can the authors use eBC vertical measurements to determine the MH if eBC emissions can occur locally along the slope? The bias between eBC-MH and T-MH is actually quite high. Moreover, in Figure 3 it can be seen that eBC concentrations during inversions are higher along the hill than in Retje (pag. 16, lines 404-406).*

**Response:**

The MHs presented in Table 5 are the same as in Figure S13. Please note that the MH reported in Table 5 represents the mean and standard deviation of 46 runs. The final results were rounded to 5 metres.

We understand the concern and we point out the impact of sources on measurements along the slopes in lines 300¬—303 in the manuscript (»It should be noted that local emission sources, slope winds and surface are expected to have a greater impact on measurements performed along the slopes as opposed to the stated standard vertical measurements.«). Nevertheless, with respect to the available options, we estimate the selection of measurements up to the solitary hill Tabor, as the most relevant and the closest to "standard" vertical measurements. As we explain below in more detail, with measurements along the circular fixed route, and therefore, passing different microenvironments of the hollow, we have reduced the direct effects of local emission sources. Certainly more, as if we measured up to the top of the hollow's north side rim, and hence, covered only the populated south-facing slopes of the hollow. Also, with measurements along the other unpopulated north-facing site of the hollow, we would not obtain the relevant vertical profile. This side of the hollow is covered with forest, has different terrain configuration and, moreover, would not provide sufficient answers in relation to the focus of our study (local air quality).

The difference between group means is indeed statistically significant, by about 0.3 of standard deviations lower eBC-MH compared to θ-MH. Population value of a slope between the θ-MH and eBC-MH, is likely to fall between 0.52 and 0.92 (95 % confidence interval). The relatively wide interval suggests a high degree of uncertainty of our method. Yet, we cannot conclude that an effect is important because the p-value, from which we determine significance, is affected by a small sample size (22 runs with both mobile temperature and eBC measurements). From frequency distribution of the θ-MH and eBC-MH data shown in the subplots of upgraded Fig. S13 in the Supplement (blue histogram for the θ-MH and gray for the eBC-MH) it is evident that there is a peak in the number of runs with θ-MH of 60 m, while for the eBC-MH, most of the runs have MH of 40 and 70 m. The former eBC-MH represents predominantly early evening runs and the latter morning runs.

Despite the method shortcomings, the difference between the θ-MH and aerosol-MH over complex terrain was observed in some other studies as well (e. g. De Wekker et al., 2004; Ferrero et al., 2010; Lang, Gohm, Wagner, 2015). Furthermore, Seibert et al. (2000) states that there is a lower agreement between the two methods for very low MH. Comparison with values of correlation coefficient and regression slope reported by e. g. Ferrero et al. (2010, 2011) show that the degree of correlation for our study is lower. However, the correlation coefficient agrees well with the one reported for the profile with timing and resolution comparability issues (Ferrero et al., 2011). Besides, root mean squared error (RMSE) indicates that our model misses actual MH values by less than 12 m (RMSE=11.43 m), which is approximately from more than 10 to 65 m less than reported in Ferrero et al. (2010, 2011). Mean absolute percentage error (MAPE) of the linear model's predictions are, on average 12.3% off from actual value. Moreover, there are more negative errors than positive, indicating a systematical underestimation of the eBC-MH. Considering absolute and relative deviations of the MHs for every run separately, the highest discrepancy was observed for the 171229 A and for 171216 C. The latter being listed under the Fig. S13 in the Supplement as a special case

The estimated uncertainty of the method used in our study is mainly result of:

1. Different timing and resolution of the θ and eBC measurements.

2. Lower comparability of the two methods during weak inversions and during mixing heights with stratified PBL structure.

3. Influence of the route characteristics (land use and cover, position in the hollow, terrain configuration and thus local flows) combined with activity of the emission sources (RWC, morning traffic).

As we state in lines 306–308 in the manuscript, resolution of temperature measurements was 2 min, while resolution of aerosol measurements was 10 s. This leads to the reduced accuracy of θ-MH. Namely, in 2 min time approximately 12.5 m distance and minimum 2 and maximum 20 m relative height had been walked from Hrib up to Tabor hill (13.5 m on average). This indicates that the θ-MH error could be as high as 20 m. Nonetheless, this is still in line with a recommended resolution range for MHs lower than 250 m (Seibert et al., 2000). This ensures that relative uncertainties do not exceed 20 %. Here, it should also be mentioned that temperature sensor time was not synchronised with the time of aerosol measurements, which might, to a lesser extent, attribute to time difference as well.

Secondly, we found that the relative difference between the two methods increases up to 54 % in case of weak temperature inversion. While during pronounced temperature inversions, the relative difference is much lower, around 15 %. This is due to less significant differences in the vertical profile during temperature inversions with near-neutral θ gradient, which makes identification of the MH more difficult. The comparability of the two methods differs between different run times as well. Overall, the agreement is better for the morning and afternoon temperature inversion runs than for the early evening runs. Higher discrepancy for the early evening runs could be the result of stratified nocturnal PBL with a stable surface boundary layer (SBL) formed at its bottom part. This is consistent with findings of studies described in Seibert et al. (2000), by Gregorič et al. (2020) and Ferrero et al. (2012).

Thirdly, an important and already mentioned contribution to the uncertainty of the method arises from the route specific measurements. Activity of emission sources along the fixed route, mainly residential wood burning (RWC) and also morning traffic, have an impact on the obtained vertical eBC profile and thus, on the determined MHs. In addition, the influence of land use and cover, the terrain configuration with related slope winds have a stronger impact as if we would use "standard" vertical measurement method. In particular, the most often occurring morning MH of 70 m, is very likely impacted by local factors. Namely, the height corresponds to the drainage area at this part of the hollow with the influence of locally induced flows. Moreover, above this height there are much less emission sources than in lower parts. These factors overlap and are interconnected, hence, the determination of various contributions is not possible.

A limitation of our method that we need to mention as well, is a time resolution of the obtained MH. Instead of recommended 1 h or less (Seibert et al., 2000), MHs in the study have a time resolution of 1.5 h (duration of the mobile measurements run). Hence, the evolution of the MH in the morning and in the early evening might not be adequately described.

With the following steps we tried to address some of the listed challenges arising from vertical mobile measurements along the slopes:

Firstly, single events, such as construction work, chimney plume or passing a heavy-duty vehicle were minimized by averaging 1-s raw data of AE51s to 10-s medians as suggested in e. g. Peters et al., 2013; Alas et al., 2018; 2019a. With the latter, the influence of AE51s measurement noise was reduced. As shown by Alas et al. (2021), however, it could still present an issue, particularly in areas with eBC levels smaller than 4 μg m$^{-3}$. However, the vertical eBC profiles presented in the paper are an average of many measurements rounded to the nearest 5 m relative height. Hence, the effect of the noise on the reported mass concentrations is further reduced. The effect of data aggregation is demonstrated by the standard deviation of the vertical eBC mass concentrations in Fig. S13. It is evident that measurements in areas with a higher concentration level have a higher variability than in the lower levels. Moreover, the special single events, which could have biased the results, were noted in the mobile measurements log-book and taken into consideration.

Secondly, with a circular fixed route, the obtained vertical profile of the hollow is a result of data points of different parts along the hollow. There is an overlap between different line segments and thus, up to 60 m relative heights, the profile is not route part specific (see Fig. S11 for the route parts height ranges). However, from there to the top of the hill at 115 m height, there is only one line segment (part of H and TR). With crossing the latter at least twice per every run (up and down the hill and after completion of some runs once more) we averaged out the single events and obtained representative distribution of eBC concentrations for the south east side of the studied area. Yet, we cannot claim that the results are valid for the other unpopulated south side of the hollow. Even if we had data from the other, unpopulated south west side of the Tabor hill, comparison and thus, the estimate of direct effects of local emission sources would not be sufficient, due to different terrain characteristics.

Lastly, for the MH determination, we have used data from the fixed stations as well. Besides θ, we considered RH measured at the three stations (Retje village station, meteorological station Hrib and Tabor background station). A significant reduction in RH among those stations was used as a sign for the height of a mixing layer. Additional control parameter, whether the MH is below or above the Tabor hill, was a comparison between eBC mass concentrations at the Retje village station and at the rural background station on Tabor hill. As shown also in the study of Gregorič et al. (2020), higher concentrations in terrain depression than on top of hill indicate MH below the altitude of the hill site.

We are aware of the limitations of the MH method used in the study. However, we estimate it meets the requirements of our study's purpose. For safe interpretation additional measurements would be needed, i. e. traffic counts, one of established vertical profile methods, wind measurements and/or pseudo-vertical measurements along the other parts of the hollow. This would, most probably, improve the mismatch between the two methods.

Alas, H. D., Stöcker, A., Umlauf, N., Senaweera, O., Pfeifer, S., Greven, S., Wiedensohler, A.: Pedestrian exposure to black carbon and PM2.5 emissions in urban hot spots: new findings using mobile measurement techniques and flexible Bayesian regression models, Journal of Exposure Science & Environmental Epidemiology, December 2020, 1–11. doi:10.1038/s41370-021-00379-5, 2021.

Gregorič, A., Drinovec, L., Ježek, I., Vaupotič, J., Lenarčič, M., Grauf, D., Wang, L., Mole, M., Stanič, S., Močnik, G.: The determination of highly time resolved and source separated black carbon emission rates using radon as a tracer of atmospheric dynamics, Atmospheric Chemistry and Physics, 20, 14139–14162, doi: https://doi.org/10.5194/acp-20-14139-2020, 2020.

**Changes in the Supplement:**

*Upgraded Fig. S12. Frequency distribution of the MHs and identification of time of day (color-coded points) added to the Figure.*

[Figure]

**Figure S12: Correlation between θ-MH and eBC-MH for temperature inversion runs in December.**

- **Comments to section 2.3.1:**

*- In this paragraph (and in many other part of the manuscript) the term "temperature"
is used instead of "potential temperature". Please, change the text accordingly.*

**Response:**

Thank you for the comment. We have changed the manuscript according to your suggestion.

*- The authors may want to cite a paper were the first derivative of the potential
temperature vertical profiles was used to study how the unstable, stable or neutral
atmospheric conditions of the atmosphere alter the distribution of aerosol backscatter
($\propto$ PM concentration) with height (Pandolfi, M., Martucci, G., Querol, X., Alastuey, A., Wilsenack,
F., Frey, S., O'Dowd, C. D., and Dall'Osto, M.: Continuous atmospheric boundary layer observations in the
coastal urban area of Barcelona during SAPUSS, Atmos. Chem. Phys., 13, 4983–4996,
https://doi.org/10.5194/acp-13-4983-2013, 2013).*

**Response:**

Thank you for the suggested paper. We have added the reference in the manuscript.

**Changes in the manuscript:**

Line 265–267: »The ratio of temperature difference between stations at different heights ($\Delta\theta/\Delta Z$) is a good
indicator of the stability of the boundary layer during temperature inversions as shown in, for instance, Petkovšek,
1978; Whiteman et al., 1999, 2014; Pandolfi et al., 2013; Holmes et al., 2015; Largeron and Staquet, 2016.«

*- I do not agree with the nomenclature used. I agree with the authors if they classify a
run as "stable atmosphere" when the potential temperature increased with altitude
along the depression. However, the authors defined the atmosphere as "mixed" when
the potential temperature decreased (negative gradient) with height. In the presence
of a negative gradient the atmosphere should be defined as "unstable" and not as
"mixed". By definition a mixed atmosphere defines a region (the mixing layer) where
the potential temperature is constant with height ($d\theta/dz = 0$). However, looking at
Figure 2, the periods between two inversion periods (shaded areas) are characterized
by equal (or very similar) potential temperature values at Tabor and Retje, thus
indicating actually a mixed atmosphere (i.e. with $d\theta/dz = 0$).*

**Reponse:**

Thank you for this comment and explanation. We have taken into account the proposed amendment and make the
change in the manuscript ("mixed" was changed to "unstable").

**Comments to section 2.3.2:**

*- The authors defined the height that separates the boundary layer from the free troposphere (FT) as MH (mixing height) irrespective of the time of the day. I would rather use, for example, PBLH (planetary boundary layer height) as a more general definition of the height separating PBL and FT as MH is usually referred as the PBLH when the atmosphere is well mixed (midday).*

**Response:**

We understand the rationale behind this. We have checked the terminology again and we agree that more concise description of terminology is needed. However, it has to be noted, that the actual PBL height could be higher than the vertical extent of our measurements. Therefore, we think, that the term PBLH is not the most suitable in our case. If the "mixing height" is derived from the eBC vertical gradient, it actually reflects the height, to where the surface pollutants are well mixed. For the purpose of simplicity and according to terminology used in Seibert et al. (2000), Ferrero et al. (2010; 2011), we propose to keep the term "mixing height" with additional explanation provided in the Section 2.3.2.

**Changes in the manuscript:**

**2.3.2 Determination of the mixing height during temperature inversions**

Vertical eBC profiles allow the determination of the height up to which the surface pollutants are well mixed (Ferrero et al., 2010 and the references therein). This atmospheric boundary layer height (ABL) is also called the mixing height (MH). It indicates pollutant accumulation in the planet boundary layer (PBL), which is the layer of the atmosphere directly influenced by the Earth's surface. The MH is related to meteorological parameters and surface roughness which are governing the behaviour of the PBL (Seibert et al., 2000). During the day with fair weather, the MH represents the height of the convective boundary layer (CBL) decoupled from the free atmosphere with a strong temperature inversion. With nocturnal radiative cooling, a statically stable boundary layer (SBL) forms at the bottom of the CBL. Above the SBL, a statically neutral layer from a previous day remains, called residual layer (RI) (Stull, 2017). Hereinafter referred MH follows the definition presented by Seibert et al. (2000) and Ferrero et al. (2011). It presents a layer affected by local emissions, regardless of the time of day. Its height may be lower than a depth of PBL, especially over complex terrain.

Previous studies (Kim et al., 2007; Angelini et al., 2009; Ferrero et al., 2010; 2011; Ferrero et al., 2016) demonstrated the accuracy of particle and eBC derived MH method, namely, atmospheric particles indicate the atmospheric dispersion state. Particles in the mixing layer accumulate due to a very stable atmosphere. Therefore, a significant difference in concentration levels between the surface boundary layer compared to the free atmosphere commonly occurs (Emeis et al., 2008; Summa et al., 2013). This sharp decrease in concentrations is defined as a mixing height (MH) (Balsley et al., 2006; Kim et al., 2007; Yang et al., 2017).

Mobile measurements of eBC and temperature along the hollow were used for the determination of mixing layer heights during the selected temperature inversion episodes. MH was calculated by means of the gradient method from 5 m averaged eBC pseudo-vertical profiles (some details on spatial averaging are provided in Sect. 2.4). The profiles were named pseudo-vertical as they were obtained with mobile measurements along the slopes of the hollow and not with commonly used methods for vertical profile measurements, e.g. radio-soundings, balloons, UAVs or lidars. It should be noted that local emission sources, slope winds and surface are expected to have a greater impact on measurements performed along slopes as opposed to the stated standard vertical measurements.

The strongest gradient of the eBC concentration profile was chosen as the mixing height. To confirm the reliability of eBC-derived MH, mixing heights for the selected temperature inversion episodes in December were determined by mobile temperature measurements as well. As with the eBC-derived MH, MH estimated from temperature profiles was determined as the height of sudden change in temperature. Comparing the results shown in the Supplement Fig. S13 it is important to consider that time resolution of temperature measurements was lower (2 min) than the resolution of particle measurements (10 s) and this can influence the accuracy of the mixing layer height determination. The agreement between different methods is very good for very pronounced inversions.

*- Given that the estimation of the eBC-MH could be hampered by eBC emissions along the slope, I would suggest the authors to compare eBC-MH also with AH-MH (AH =*

*absolute humidity) during mobile run measurements.*
*- My suggestion is to include in this work the AH too (of course if RH measurements are*
*available during mobile measurements).*

**Response:**

Unfortunately, we were not measuring RH with the mobile unit, only at the measuring stations. Due to this limitation we added only the RH measured at the stations in the Figure S13.

**Changes in the Supplement:** Figure S13. See answers and changes above.

- **Comments to Figure 2:**
*- As in my previous comment. Looking at Figure 2, the periods between two*
*temperature inversion periods (shaded areas) were characterized by equal (or very*
*similar) potential temperature values at Tabor and Retje, thus indicating actually a*
*mixed atmosphere (i.e. with dθ/dz = 0). However, in the main text the authors defined*
*the atmosphere during these periods as periods with dθ/dz < 0 (whereas it seems from*
*Figure 2 that dθ/dz = 0). So, I have not understood how the authors defined a "mixed"*
*atmosphere: when dθ/dz = 0 or when dθ/dz < 0?*

**Response:**

Thank you for this comment. In Figure 2 we have marked only the longer temperature inversion periods (lasting a day or more), while we did not specifically mark temperature inversions that last only part of the day and periods with mixed atmosphere. Thus, periods between the shaded areas represents the days with more dynamical weather, yet include some shorter periods with temperature inversion or isothermal conditions as well. However, for further analysis, the criteria for the temperature inversion was dθ/dz > 0 and for the mixed atmosphere dθ/dz < 0. We have enlarged the plot since from the previous one, the difference between the stations was not clearly visible.

*- Interestingly, no rain was observed during temperature inversion periods. Very often,*
*rain was detected just before and just after the inversion periods (shaded areas in*
*Figure 2). Any comment about this?*

**Response:**
Radiation temperature inversions were studied as problematic cases when the concentration of local emissions increased. During precipitation periods, the radiative inversion was very quickly dissipated by advection or even turbulence. The shallow depression of Loški Potok is located on an elevated terrain with a relatively large sky view factor, thus the wind over the depression can easily enter the inversion layer and mixes the air rapidly. Therefore, the problematic cases of air quality are always related with radiation inversion with clear and calm nocturnal weather without any precipitation.

*- During stable atmospheric conditions the eBC concentrations were higher than usual*
*at the top of the hill too (Tabor site). Was this relative increase due to some eBC*
*transport from below (e.g. Retje)? or was due to eBC sources along the hill*
*accumulated during stable conditions?*

**Response:**

We discuss this in more detail in the paper Glojek et al., 2020. According to the diurnal profile of eBC at the Tabor site presented in the mentioned study, the relative increase in eBC during temperature inversions at the site is mainly the result of sources in the village of Hrib, which most often lies above the inversion. However, with thermal forcing due to increased solar irradiation in the middle of the day, transport of pollution from the Retje (the bottom of the depression) to Tabor was occasionally observed as well.

Glojek, K., Gregorič, A., Močnik, G., Cuesta-Mosquera, A., Wiedensohler, A., Drinovec, L. and Ogrin, M.: Hidden black carbon air pollution in hilly rural areas—a case study of Dinaric depression, Eur. J. Geogr., 11(2), 105–122, doi:10.48088/ejg.k.glo.11.2.105.122, 2020.

**REVIEWER 2:**

*This paper presents an analysis of the impact of ground temperature inversions on total particle mass concentrations and in particular on black carbon. The experiments were carried out in rural Slovenia, with emphasis was on wood combustion aerosol pollution. Overall, the paper is well written and the experimental approach is well described. I would recommend its publication in ACP after some revisions:*

**Comments:**

- How is the profile along the fixed route? Why is only the profile perpendicular to the route shown?

**Response:**

Thank you for the question, we have included the horizontal gradients in the manuscript, under the Table 4 of 3.2 Chapter.

**Changes in the manuscript:**

At the end of the chapter 3.2:

A deeper analysis of the pollutants' spatial distributions during temperature inversions is shown in Figure 4. The determined line segments along the fixed route inside the Retje hollow are listed and described in Table 4 with their spatial locations marked in Figure 5. Here, the goal was to show horizontal variability of eBC and $PM_{2.5}$ along the hollow with respect to different times of the day. To this end, line segments at similar relative heights were selected (a maximum difference of 30 m per route segment; see Table 4).

[Figure]

**Figure 4: Variability of eBC (top panels) and $PM_{2.5}$ (bottom panels) mass concentrations during morning (6:30–8:30 LT, red line), afternoon (12.00:14.00 LT, yellow line) and evening (17:00–19:00 LT, purple line) winter-time temperature inversion along different parts of the fixed route in the Retje hollow.**

**Table 5: List and description of the different route segments along the Retje hollow.**

| Location code | Descriptions | Altitudes (m) | eBC/ $PM_{2.5}$ |
|---|---|---|---|
| **VB–BR** | Bottom route in a village Retje-Hrib direction. | 705.9–741.5 | 0.12 |
| **VB–C** | Line segment in a village Retje-chapel st. Florjan direction. | 706.7–728.9 | 0.11 |
| **VB–VS** | Bottom part of village Retje in a west-north direction. | 707.1–719.2 | 0.07 |
| **VS & SR** | Slope route in a village Retje-Hrib direction. | 720.3–758.4 | 0.16 |

[Figure]

**Figure 5: The location codes assigned to the line segments of the fixed route.**

eBC and $PM_{2.5}$ mass concentration variability trend differs throughout the route and it is day time dependent as well. Namely, mass concentrations of both pollutants are 2- to 4-times higher during evening temperature inversions as compared to inversions in the morning and in the afternoon. This applies to all locations inside the hollow. Yet, the

difference between the run times is the highest in Retje village, with up to 5- and 7-times higher evening eBC and PM$_{2.5}$ levels, respectively.

Unlike the PM$_{2.5}$, the eBC trend along the line segment VB–BR depends on the time of the day. In the morning, from 6:30 to 9:00, eBC levels with the mean value of 3 µg m$^{-3}$ remain constant along the whole bottom route (BR), whereas around noon and in the early evening, eBC mass concentrations vary. As seen with the PM$_{2.5}$ levels, early evening eBC mass concentrations increase from 6.6 µg m$^{-3}$ to 11.6 µg m$^{-3}$ towards the village of Retje. In the other direction, from the village of Retje to st. Florjan chapel (VB–C), levels of both pollutants decrease, with the highest drop observed during evening temperature inversions, particularly in PM$_{2.5}$.

Horizontal gradients of the pollutants' mass concentrations differ the most at Retje village located at the bottom of the hollow (VB–VS) and at the slope street in the direction from Retje village towards the settlement Hrib (VS & SR). This relates to morning and evening temperature inversion runs, while for the afternoon runs variability of the pollutants is similar. In the morning and in the evening, PM$_{2.5}$ mass concentrations over the whole village at the bottom of the Retje hollow are elevated and steeply decline further away from the village, especially in the evening. In the afternoon, however, greater mass concentrations were measured along the village on the slope (VS). This is seen in Figure 4, line segment VB–VS, as a sharp increase in PM$_{2.5}$ mass concentrations at a distance of 200 m away from the route segment starting point. The decrease of eBC along the slope road (VS & PC) from Retje village towards Hrib is smaller than that of PM$_{2.5}$. Moreover, during morning runs, eBC mass concentrations in the middle of the road segment increase, i. e. from 4.4 to 9.2 µg m$^{-3}$.

Line 596–598: Moreover, the high correlation between eBC and PM$_{2.5}$ mass concentrations along the whole hollow indicates homogeneous atmospheric conditions with the prevalence of one emission source. However, different morning and evening horizontal gradients of eBC and PM$_{2.5}$ were observed during temperature inversions in the village of Retje. We partially interpret this as secondary aerosol formation (SOA).

There are two Figures S12.

**Response:**

Thank you for this comment. We do see oversight and we have changed wrong Figure S12 into S11 in line with the pdf* document.

- References to Sections within the text should be checked (e.g., line 315 and 435)

**Response:**

Thank you for the noticed incorrect citations. We have checked them and corrected accordingly.

**Changes in the manuscript:**

Line 315: »Concerning the second goal, mobile measurements were spatially averaged according to the method described in Alas et. al., 2019a and Alas et al., 2019b.«

- How strongly are the eBC vertical profile influenced by direct eBC emissions? Is an error estimation possible?

The difference between group means is indeed statistically significant, by about 0.3 of standard deviations lower eBC-MH compared to θ-MH. Population value of a slope between the θ-MH and eBC-MH, is likely to fall between 0.52 and 0.92 (95 % confidence interval). The relatively wide interval suggests a high degree of uncertainty of our method. Yet, we cannot conclude that an effect is important because the p-value, from which we determine significance, is affected by a small sample size (22 runs with both mobile temperature and eBC measurements). From frequency distribution of the θ-MH and eBC-MH data shown in the subplots of upgraded Fig. S13 in the Supplement (blue histogram for the θ-MH and gray for the eBC-MH) it is evident that there is a peak in the number of runs with θ-MH of 60 m, while for the eBC-MH, most of the runs have MH of 40 and 70 m. The former eBC-MH represents predominantly early evening runs and the latter morning runs.

Despite the method shortcomings, the difference between the θ-MH and aerosol-MH over complex terrain was observed in some other studies as well (e. g. De Wekker et al., 2004; Ferrero et al., 2010; Lang, Gohm, Wagner, 2015). Furthermore, Seibert et al. (2000) states that there is a lower agreement between the two methods for very low MH. Comparison with values of correlation coefficient and regression slope reported by e. g. Ferrero et al.

(2010, 2011) show that the degree of correlation for our study is lower. However, the correlation coefficient agrees well with the one reported for the profile with timing and resolution comparability issues (Ferrero et al., 2011). Besides, root mean squared error (RMSE) indicates that our model misses actual MH values by less than 12 m (RMSE=11.43 m), which is approximately from more than 10 to 65 m less than reported in Ferrero et al. (2010, 2011). Mean absolute percentage error (MAPE) of the linear model's predictions are, on average 12.3% off from actual value. Moreover, there are more negative errors than positive, indicating a systematical underestimation of the eBC-MH. Considering absolute and relative deviations of the MHs for every run separately, the highest discrepancy was observed for the 171229 A and for 171216 C. The latter being listed under the Fig. S13 in the Supplement as a special case.

One of the major limitation of our model is that the dataset contains only a small number of different values, since the MHs were determined only during temperature inversions at the specific times in a day. In this case, also using for example, the bootstrap sampling, does not improve the sampling distributions of the variables and thus the linear regression model. As we explain below in more detail, local spatiotemporal factors influencing the eBC profiles are interconnected and thus, their separate contribution in the frame of this study is not possible. Besides, for safe interpretation, we lack of additional data (e. g. traffic counts, one of established vertical profile methods, wind measurements and/or pseudo-vertical measurements along the other parts of the hollow).

The estimated uncertainty of the method used in our study is mainly result of:

1. Different timing and resolution of the $\theta$ and eBC measurements.

2. Lower comparability of the two methods during weak inversions and during mixing heights with stratified PBL structure.

3. Influence of the route characteristics (land use and cover, position in the hollow, terrain configuration and thus local flows) combined with activity of the emission sources (RWC, morning traffic).

As we state in lines 306–308 in the manuscript, resolution of temperature measurements was 2 min, while resolution of aerosol measurements was 10 s. This leads to the reduced accuracy of $\theta$-MH. Namely, in 2 min time approximately 12.5 m distance and minimum 2 and maximum 20 m relative height had been walked from Hrib up to Tabor hill (13.5 m on average). This indicates that the $\theta$-MH error could be as high as 20 m. Nonetheless, this is still in line with a recommended resolution range for MHs lower than 250 m (Seibert et al., 2000). This ensures that relative uncertainties do not exceed 20 %. Here, it should also be mentioned that temperature sensor time was not synchronised with the time of aerosol measurements, which might, to a lesser extent, attribute to time difference as well.

Secondly, we found that the relative difference between the two methods increases up to 54 % in case of weak temperature inversion. While during pronounced temperature inversions, the relative difference is much lower, around 15 %. This is due to less significant differences in the vertical profile during temperature inversions with near-neutral $\theta$ gradient, which makes identification of the MH more difficult. The comparability of the two methods differs between different run times as well. Overall, the agreement is better for the morning and afternoon temperature inversion runs than for the early evening runs. Higher discrepancy for the early evening runs could be the result of stratified nocturnal PBL with a stable surface boundary layer (SBL) formed at its bottom part. This is consistent with findings of studies described in Seibert et al. (2000), by Gregorič et al. (2020) and Ferrero et al. (2012).

Thirdly, an important and already mentioned contribution to the uncertainty of the method arises from the route specific measurements. Activity of emission sources along the fixed route, mainly residential wood burning (RWC) and also morning traffic, have an impact on the obtained vertical eBC profile and thus, on the determined MHs. In addition, the influence of land use and cover, the terrain configuration with related slope winds have a stronger impact as if we would use "standard" vertical measurement method. In particular, the most often occurring morning MH of 70 m, is very likely impacted by local factors. Namely, the height corresponds to the drainage area at this part of the hollow with the influence of locally induced flows. Moreover, above this height there are much less emission sources than in lower parts. These factors overlap and are interconnected, hence, the determination of various contributions is not possible.

A limitation of our method that we need to mention as well, is a time resolution of the obtained MH. Instead of recommended 1 h or less (Seibert et al., 2000), MHs in the study have a time resolution of 1.5 h (duration of the mobile measurements run). Hence, the evolution of the MH in the morning and in the early evening might not be adequately described.

With the following steps we tried to address some of the listed challenges arising from vertical mobile measurements along the slopes:

Firstly, single events, such as construction work, chimney plume or passing a heavy-duty vehicle were minimized by averaging 1-s raw data of AE51s to 10-s medians as suggested in e. g. Peters et al., 2013; Alas et al., 2018; 2019a. With the latter, the influence of AE51s measurement noise was reduced. As shown by Alas et al. (2021), however, it could still present an issue, particularly in areas with eBC levels smaller than 4 μg m-3. However, the vertical eBC profiles presented in the paper are an average of many measurements rounded to the nearest 5 m relative height. Hence, the effect of the noise on the reported mass concentrations is further reduced. The effect of data aggregation is demonstrated by the standard deviation of the vertical eBC mass concentrations in Fig. S13. It is evident that measurements in areas with a higher concentration level have a higher variability than in the lower levels. Moreover, the special single events, which could have biased the results, were noted in the mobile measurements log-book and taken into consideration.

Secondly, with a circular fixed route, the obtained vertical profile of the hollow is a result of data points of different parts along the hollow. There is an overlap between different line segments and thus, up to 60 m relative heights, the profile is not route part specific (see Fig. S11 for the route parts height ranges). However, from there to the top of the hill at 115 m height, there is only one line segment (part of H and TR). With crossing the latter at least twice per every run (up and down the hill and after completion of some runs once more) we averaged out the single events and obtained representative distribution of eBC concentrations for the south east side of the studied area. Yet, we cannot claim that the results are valid for the other unpopulated south side of the hollow. Even if we had data from the other, unpopulated south west side of the Tabor hill, comparison and thus, the estimate of direct effects of local emission sources would not be sufficient, due to different terrain characteristics.

Lastly, for the MH determination, we have used data from the fixed stations as well. Besides θ, we considered RH measured at the three stations (Retje village station, meteorological station Hrib and Tabor background station). A significant reduction in RH among those stations was used as a sign for the height of a mixing layer. Additional control parameter, whether the MH is below or above the Tabor hill, was a comparison between eBC mass concentrations at the Retje village station and at the rural background station on Tabor hill. As shown also in the study of Gregorič et al. (2020), higher concentrations in terrain depression than on top of hill indicate MH below the altitude of the hill site.

Alas, H. D., Stöcker, A., Umlauf, N., Senaweera, O., Pfeifer, S., Greven, S., Wiedensohler, A.: Pedestrian exposure to black carbon and PM2.5 emissions in urban hot spots: new findings using mobile measurement techniques and flexible Bayesian regression models, Journal of Exposure Science & Environmental Epidemiology, December 2020, 1–11. doi:10.1038/s41370-021-00379-5, 2021.

Gregorič, A., Drinovec, L., Ježek, I., Vaupotič, J., Lenarčič, M., Grauf, D., Wang, L., Mole, M., Stanič, S., Močnik, G.: The determination of highly time resolved and source separated black carbon emission rates using radon as a tracer of atmospheric dynamics, Atmospheric Chemistry and Physics, 20, 14139–14162, doi: https://doi.org/10.5194/acp-20-14139-2020, 2020.

**Changes in the Supplement:**

Upgraded Fig. S12. Frequency distribution of the MHs and identification of time of day (color-coded points) added to the Figure.

[Figure]

**Figure S12: Correlation between θ-MH and eBC-MH for temperature inversion runs in December.**

- BC source apportionment reported in the cited manuscripts still show significant traffic contribution during certain periods during the rush hour period. Would be interesting to have some numbers also in the present manuscript together with a comment, e.g., line 585 what is the contribution of local morning traffic to total eBC?

**Changes in the manuscript:**

Lines 585–588: »Local morning traffic (people driving to a local school and to their workplace) contribute slightly as well, especially when it comes to eBC mass concentration levels (Glojek et al., 2020). During the peak hour, at 9:00, traffic accounts for 34 % of eBC in Retje (Glojek et al., 2020).«

---

## Author Response (AR2)

REVIEWER Comment to the manuscript:

**"The limitations of the proposed technique (eBC-MH) should be better described in the paper. Just please summarize in the manuscript your response to my comment "Comments to Table 5, Figure 5 and related text".**
Thank you for the comment and suggestion. We summarised our response to the comment in added sub-section 3.3.1 (Limitations of the mixing height determination) under Results. The text in blue presents the new text in the manuscript.

**Changes in the manuscript:**
*Line 532–587:"*

[revised manuscript text omitted]